# Single cell RNA sequencing of human liver reveals distinct intrahepatic macrophage populations

Sonya A. MacParland et al.[#]

The liver is the largest solid organ in the body and is critical for metabolic and immune functions. However, little is known about the cells that make up the human liver and its immune microenvironment. Here we report a map of the cellular landscape of the human liver using single-cell RNA sequencing. We provide the transcriptional profiles of 8444 parenchymal and non-parenchymal cells obtained from the fractionation of fresh hepatic tissue from five human livers. Using gene expression patterns, flow cytometry, and immunohistochemical examinations, we identify 20 discrete cell populations of hepatocytes, endothelial cells, cholangiocytes, hepatic stellate cells, B cells, conventional and non-conventional T cells, NK-like cells, and distinct intrahepatic monocyte/macrophage populations. Together, our study presents a comprehensive view of the human liver at single-cell resolution that outlines the characteristics of resident cells in the liver, and in particular provides a map of the human hepatic immune microenvironment.

Correspondence and requests for materials should be addressed to S.A.M. (email: s.macparland@utoronto.ca) or to G.D.B. (email: gary.bader@utoronto.ca) or to I.D.M. (email: Ian.McGilvray@uhn.ca). [#]A full list of authors and their affliations appears at the end of the paper.

The liver is vital for human metabolism and immune function. A reference map of the healthy human liver landscape at single-cell resolution is critical to understanding the pathogenesis and treatment of liver disease. This landscape has been difficult to describe[1], mainly because fresh human liver tissue access is scarce and the tissue is difficult to fractionate without damaging fragile resident cell populations. One approach to creating an unbiased map of the human liver cellular landscape is to combine careful dissociation of relatively large segments of fresh, healthy human liver with single-cell RNA sequencing (scRNA-seq). Although scRNA-seq is a powerful tool for describing highly heterogeneous cell populations such as those found in whole tissue[2,3], it has not yet been widely applied to describe whole human organs, with only maps of isolated islet cells from the human pancreas published until now[4–11]. At present, the only single-cell transcriptomic map for the whole liver is from mice[12].

The current understanding of human liver cellular organization is based on the building block of the hepatic acinus. The acinus consists of portal triads, each comprised of a hepatic artery, portal vein, and bile duct, hepatocytes and the biliary tree that radiate outward and are sandwiched between a capillary network and a central draining hepatic vein. The bulk of the hepatic acinus consists of cords of hepatocytes arranged back to back and sandwiched between liver sinusoidal endothelial cells (LSECs). Running between the hepatocytes are fine biliary ducts that drain outwards into the portal triad bile duct, while blood drains inwards towards the central veins. Within the acinus are parenchymal cells (hepatocytes) and non-parenchymal cells (NPCs) (cholangiocytes, endothelial cells, Kupffer cells (KCs)), hepatic stellate cells and liver resident, and infiltrating lymphocytes—including B cells, conventional, and non-conventional T cells (including ILCs, NKT cells, and MAIT cells) and natural killer (NK) cells. Liver immune cells are distributed in specific patterns, though many details remain unknown in terms of cellular location and cellular phenotypes. For example, there are few direct examinations of human KCs, even though they represent the large majority of the body's macrophages[1].

Here we apply liver tissue dissociation techniques we previously developed[13,14] to perform an unbiased examination of the cellular landscape of the normal human liver via scRNA-seq. We identify 20 hepatic cell populations from the transcriptional profiling of 8444 cells obtained from liver grafts of five healthy neurologically deceased donors (NDD). By examining the most differentially expressed (DE) genes of each cluster, and using known landmark genes or characterizing markers known from cell-specific gene expression, flow cytometry, or immunohistochemical examinations of human liver tissue, we find distinct populations of hepatocytes, endothelial cells, cholangiocytes, hepatic stellate cells, KCs, B cells, conventional and non-conventional T cells, and NK cells. These evaluations uncover aspects of the immunobiology of the liver, including the presence of two distinct populations of liver resident macrophages with inflammatory and non-inflammatory/immunoregulatory functions. This transcriptomic map serves as a fundamental baseline description of the human liver.

## Results

### A protocol for human liver dissociation for scRNA-seq. A central problem in liver tissue dissociation is that hepatocytes and cholangiocytes are sensitive to cell death, and other cells activate, in response to cell manipulation[15–17]. We developed a cell isolation approach (Fig. 1a, b) without density gradient or column purification, or flow cytometry. We found that cell sorting approaches led to loss of fragile cells, particularly hepatocytes and

cholangiocytes, and the relative enrichment of more robust NPCs, such as endothelial cells/KCs. Preliminary scRNA-seq experiments showed an under-representation of NPC cell populations in the sorted NPC-enriched fraction compared to total liver homogenate (TLH) (Supplementary Fig. 1). Specifically, the NPC fraction contained two endothelial cell clusters compared to three in the TLH. Furthermore, when we targeted 6000 viable cells for scRNA-seq from TLH and NPC fractions, the NPC fraction yielded far fewer viable cells than the TLH (Supplementary Fig. 1) indicating liver cell sensitivity to flow sorting.

Using our liver dissociation protocol, cell viability of the TLHs obtained from five caudate lobes ranged from 49 to 90% viable by trypan blue exclusion (Supplementary Fig. 2a). However, the actual number of cells profiled per caudate after filtering for library size and mitochondrial transcript percentage (Fig. 2a) ranged from 1073 to 3255 cells per sample (between 17.9 and 54.3% passing quality control). The mean library size (number of UMIs detected/cell) for TLHs was 5227 (range 3122–6043 UMIs) and the mean number of genes detected per cell was 1313 (range 906–1537 genes) (Supplementary Figs. 2, 3). The variation in numbers of cells profiled may be attributed to differences in sample viability and technical differences in cell capture rates for each scRNA-seq run. Standard scRNA-seq filtering excludes cells with a high ratio of reads from mitochondrial genome transcripts, indicating potential plasma membrane rupture and dissociation-based damage[18]. This filter is often set at 10%, but hepatocytes can have very high mitochondrial content[19], thus we chose a threshold of 50% to optimize keeping hepatocytes and removing dead and dying cells. Evaluation of six mitochondrial cut-offs ranging from 10–60% (Supplementary Figs. 4, 5) showed that all clusters (except cluster #6 at the 10% cutoff) are identified as unique populations in t-SNE plots at all cut-offs, indicating that our map is robust to this threshold. As expected, hepatocytes were most susceptible to removal by this filter (Fig. 3), while endothelial cells, macrophages were consistently represented in all livers profiled. Cell doublets are normally detected in scRNA-seq experiments as cells with abnormally high library size. However, we did not detect a natural threshold in library size per cell that we could select to remove doublets. Further, there are naturally occurring binucleated hepatocytes that are expected to have a high library size, which would make it difficult to distinguish between doublets and binucleated cells. A doublet filter set to remove cells with the top 99.9% library size (following standard protocols[20]), mostly removed hepatocyte (cluster #14) and plasma cell (cluster #7) populations. As both binucleated hepatocytes[21] and tissue trafficking plasmablasts[22] have been previously described, we suspect that the cells identified as "doublets" may be single biological cells containing two nuclei and thus did not apply doublet filtering to our map (Supplementary Fig. 6).

### The landscape of cells in the healthy human liver. We applied our liver single-cell dissociation protocol and scRNA-seq technology to identify resident liver cells in TLHs from NDD liver transplant donors (donor characteristics found in Supplementary Table 1). Importantly, NDD grafts are subjected to the systemic inflammation that accompanies brain death and are thus themselves mildly inflamed[23]. However, we confirmed normal histological patterns in these livers (Supplementary Fig. 7). Pooling results from all donors, we captured 8444 cells (after filtering out low viability cells) that clustered into 20 discrete cell populations (Fig. 2c, f, Supplementary Data 1, 2) that are described below.

### Hepatocytes. Hepatocytes are the building blocks of the liver and play a role in detoxification, lipolysis, and gluconeogenesis[24]. In

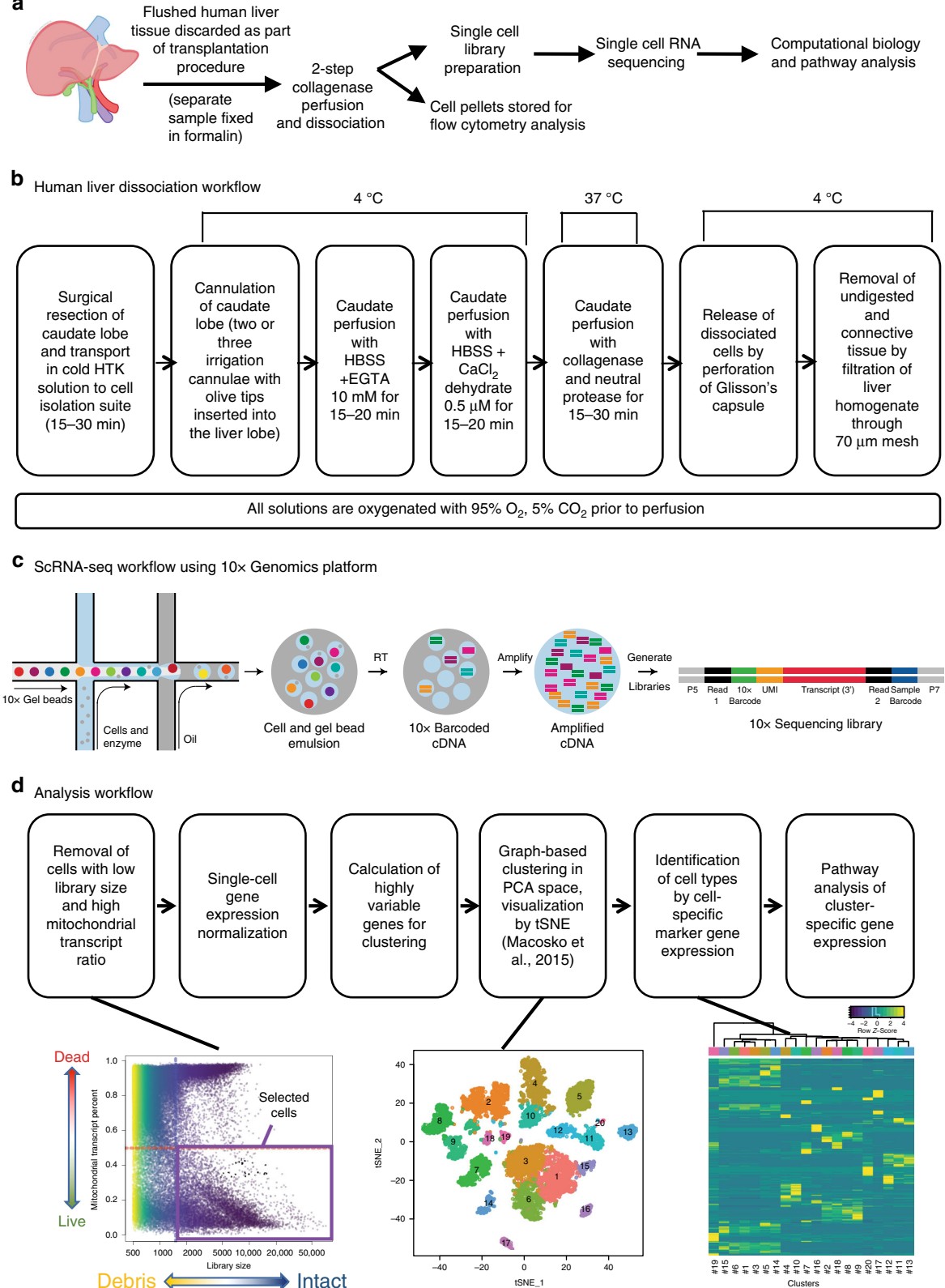

**Fig. 1** ScRNA-seq profiling of parenchymal and non-parenchymal cells from the human liver. **a** Overview of the single-cell isolation and analysis workflow. Workflows for **b** the dissociation of human caudates to single-cell homogenates, **c** the generation of scRNA-seq cDNA expression libraries using the 10x Genomics genomics platform and, **d** data analysis

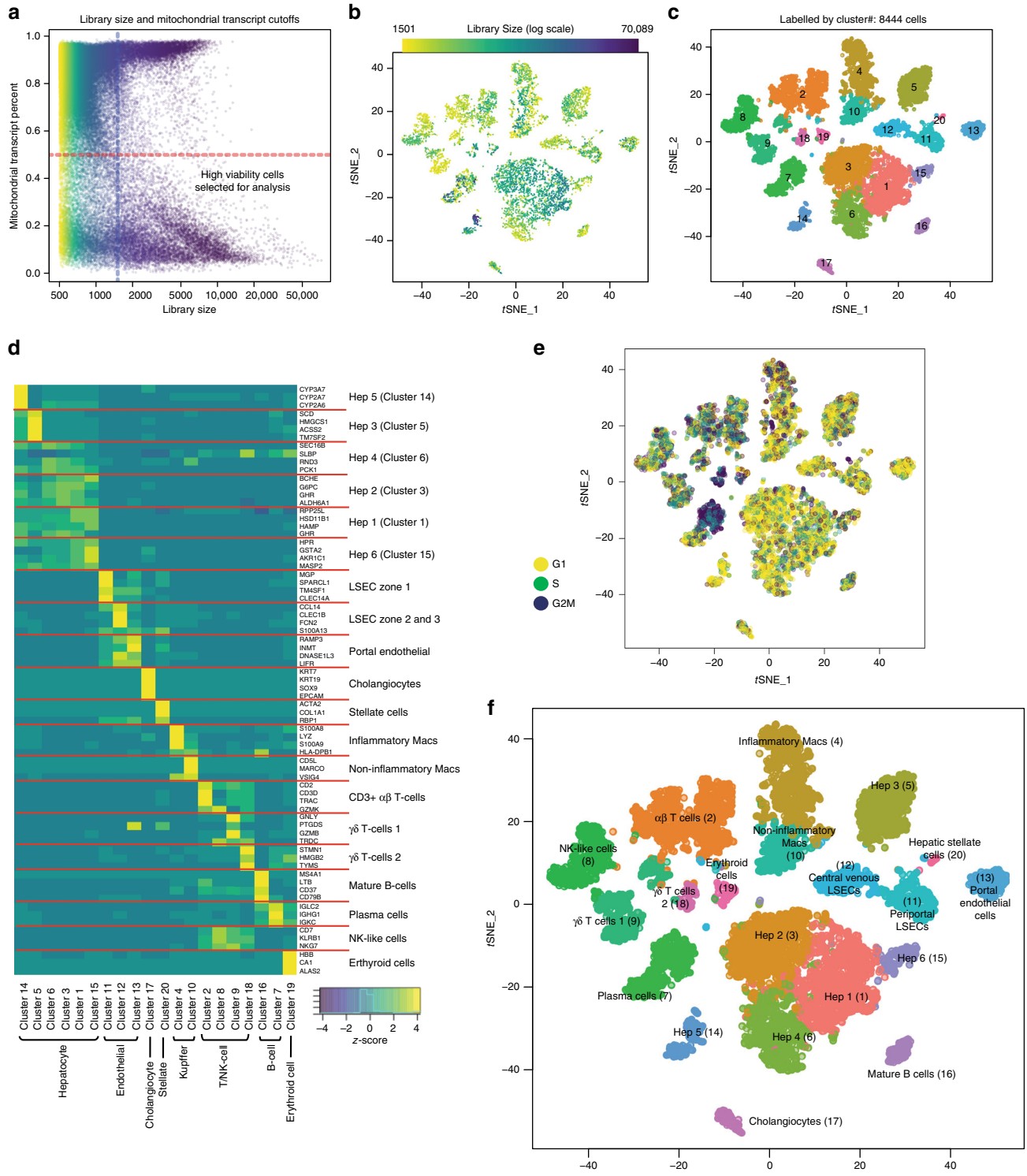

**Fig. 2** 20 distinct cell populations were revealed in healthy human livers. **a** Viable cells were identified from the single-cell libraries having a minimum library size of 1500 transcripts and a maximum of 50% mitochondrial transcript proportion. **b** *t*-SNE projection of 8444 liver cells (each point represents a single cell). Cells are colored by library size, with darker colors indicating larger libraries. **c** *t*-SNE projection where cells that share similar transcriptome profiles are grouped by colors representing unsupervised clustering results. **d** Heat map analysis using known gene expression profiles of hepatocytes/ immune cells. The identity of each cluster was assigned by matching the cluster expression profile with established cell-specific marker gene expression for hepatocytes, endothelial cells, cholangiocytes, and immune cells. **e** Cell-cycle phase prediction showed that hepatocyte clusters were less proliferative than immune cell clusters. **f** Cluster map showing the assigned identity for each cluster defined in **c**. The cluster number of each potential cell population is indicated in parentheses. DE: differentially expressed, MACs: macrophages, PCA: principal component analysis, *t*-SNE: *t*-distributed stochastic neighbor embedding, PCs: principal components

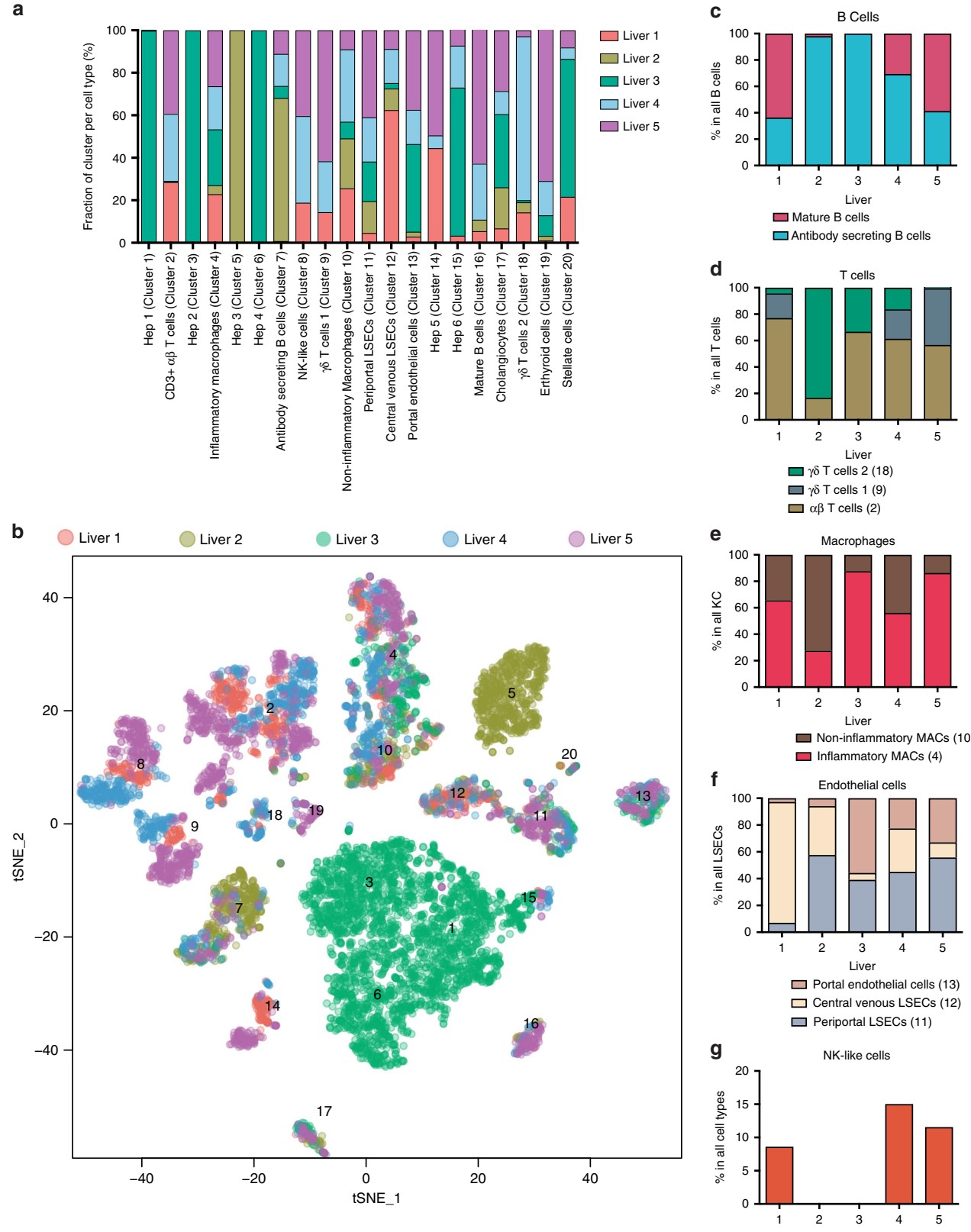

**Fig. 3** Contribution of cells to each scRNA-seq cluster by sample and subpopulation analysis. **a** The proportion of cells that contributed to each cluster by liver sample. **b** *t*-SNE projection of all cells, colored by the source donor, and labeled with cluster number. Most cell-type associated clusters are made up of multiple donors. Hepatocyte clusters, on the other hand, appear to segregate by donor. **c** Proportions of mature vs. plasma B cells across five liver samples as a percentage of total B cells. Similar subpopulation analyses were carried out for **d** αβ & γδ T cells **e** Macrophages (Macs), **f** Endothelial cells, and **g** Natural killer (NK)-like cells. LSECs liver sinusoidal endothelial cells

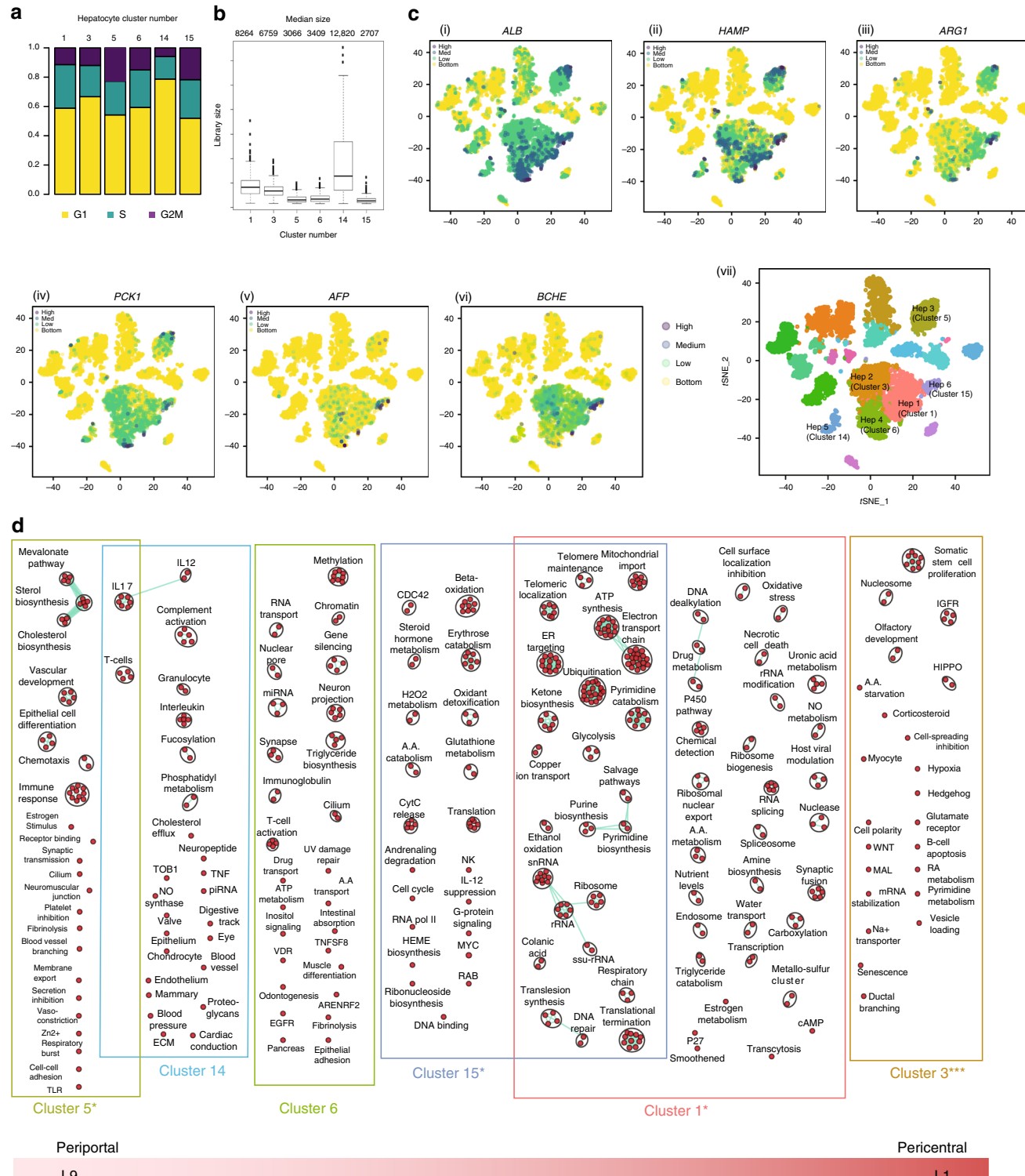

mice and human, hepatocytes have been described as having distinct functions based on their location in the hepatic acinus, known as their zonation[12,24]. We found six distinct hepatocyte populations, comprising Clusters 1, 3, 5, 6, 14, and 15 (Figs. 2f, 4ci), that were generally less proliferative cells (Figs. 2e, 4a), and showed enriched *ALB* (Albumin) expression, a hallmark of hepatocytes.

To directly infer hepatocyte cluster zonation patterns, single-molecule FISH (smFISH) and laser capture microdissection

studies are required, these techniques were not a part of this study. However, a systematic comparison was carried out examining gene expression patterns in the identified human hepatocyte clusters with respect to the zonated gene expression patterns previously shown in mouse[12] (Supplementary Fig. 8, Fig. 4d). This analysis revealed that gene expression patterns in four human hepatocyte clusters correlated significantly with zonated gene expression patterns identified in the mouse sinusoid (Supplementary Fig. 8, Supplementary Data 3). Two clusters

**Fig. 4** ScRNA-seq analysis of hepatocyte populations. **a** Distribution of hepatocytes by cell-cycle phase (G1, G2/M, S) and hepatocyte cluster (1, 3, 5, 6, 14, 15). **b** Box plot of library size for each hepatocyte cluster with median library size (top) and graphically denoted median (dark horizontal line). Outliers (black dots) and interquartile range (black box) are indicated. **c** t-SNE plots showing the expression of general hepatocyte markers based on PCA clustering of 8444 cells. **c** (i) *ALB*, **c** (ii) *HAMP*, **c** (iii) *ARG1*, **c** (iv) *PCK1*, **c** (v) *AFP*, **c** (vi) *BCHE*. Legend for relative expression of each marker from lowest expression (yellow dots) to highest expression (purple dots) (top left). **c** (vii) *t*-SNE projection showing a reference map of all six hepatocyte clusters. **d** Pathway enrichment analysis examining active cellular pathways in clusters 1, 3, 5, 6, 14 & 15. The size of the nodes represents the number of genes in a particular pathway. Highly related pathways are grouped into a theme (black circle) and labeled in Cytoscape (Version 3.6.1). Intra- and inter-pathway relationships are shown (green lines) and represent the number of genes shared between each pathway. Periportal and pericentral zones are assigned in relation to correlation analysis between mouse and human liver in Supplementary Fig. 8. Statistical significance of the correlation between mouse liver layers and human liver clusters (denoted under each pathway analysis) calculated using Pearson correlation. ***$P < 0.001$, **$P < 0.01$, *$P < 0.05$. *t*-SNE t-distributed stochastic neighbor embedding

showed weak correlation, possibly due to differences in the genes that define mouse and human zonated liver expression patterns. Specifically, human Cluster 5 correlated with the most periportal mouse liver layer, while human Cluster 3 correlated with the most central venous mouse liver layer. Human Cluster 1 was correlated with mouse layers 2 and 3 (more periportal) and Cluster 15 correlated with an interzonal mouse layer (Layer 4). Clusters 6 and 14 did not correlate significantly with any mouse layers, possibly because the top DE genes defining these clusters were not found in the top 94 genes which defined zonation in mouse livers.

To study the function of the human clusters, we applied pathway analysis to identify active cellular pathways in each cluster (Fig. 4d, Supplementary Data 4). Characteristic functional pathways in sinusoidal zones have been well described[12,24]. Using these approaches, we describe the possible zonation for human hepatocyte clusters with the expectation that natural hepatocyte heterogeneity will be better addressed as more human liver samples are profiled across multiple study sites.

Zone 1 (periportal) hepatocytes have a known role in gluconeogenesis and β-oxidation[24]. Cluster 5 showed enriched expression of genes involved in lipid and cholesterol synthesis such as *SCD* (Supplementary Fig. 9), *HMGCS1*, and *ACSS2* with enhanced expression of *PCK1* (Fig. 4c.iv) and *CPS1* (Supplementary Data 1, Supplementary Fig. 9), genes that are enriched in the periportal hepatocytes of mice[12], suggesting that this cluster may represent Zone 1 hepatocytes. A further indication of a periportal nature is enriched expression of urea cycle gene *ARG1* (Fig. 4c.iii). (Top DE genes cluster 5: *SCD, HMGCS1, ACSS2, TM7SF2, TMEM97, CP, CRP, SLPI, C2orf82, ACAT2, TM4SF5, MSMO1, LEPR*). Pathway analysis revealed that Cluster 5 was enriched for pathways characteristic of liver periportal function including cholesterol and sterol biosynthesis along with numerous active immune pathways (Fig. 4d).

In mice, Zone 3 or central venous hepatocytes play a role in drug metabolism and detoxification, constitutively expressing high levels of cytochrome P450 enzymes[12]. Clusters 3 and 1 were enriched for the expression of *BCHE* (butyrylcholinesterase), involved in drug metabolism (Cluster 3 top DE genes: *BCHE, G6PC, GHR, ALDH6A1, RCAN1, AR, RP4-710M16.2, LINC00261, PLIN1, RP11-390F4.3*) (Fig. 4c.vi.). Cluster 1 also showed enriched expression of *HAMP* (Fig. 4c.ii), a gene that is enriched in mouse midzonal hepatocytes[12]. (Cluster 1 top DE genes: *RPP25L, HSD11B1, HAMP, GHR, APOM, APOC4-APOC2, TKFC, G6PC, G0S2, PON3, C1orf53, TTC36, GOLT1A, RCAN1, RP4-710M16.2, FST, MCC, AQP9, PLIN1*). In the comparison with zonated mouse liver data, Cluster 3 correlated most significantly with the most central venous mouse sinusoid layer (Supplementary Fig 8). Cluster 1 showed a significant correlation with mouse layers 2 and 3, suggesting that Clusters 3 and 1 may be more central venous. Pathway analysis revealed that the cells making up these two clusters were active in cellular pathways characteristic of zone 3 functions in mice and human including

P450 pathways, drug metabolism, Wnt activation, hypoxia, amino acid biosynthesis, and glycolysis[12,24] (Fig. 4d), supporting the notion that these cells might have a central venous origin.

Cluster 15 had the smallest library size of the hepatocyte clusters (2707 UMIs per cell) (Fig. 4b) (Top DE genes: *HPR, GSTA2, AKR1C1, MASP2, NNT, SAA4, MRPS18C, OCIAD1, APOA5, TTR*). Gene expression patterns in this cluster correlated significantly with mouse sinusoid layer 4, suggesting that these cells may be interzonal hepatocytes (Supplementary Fig 8). Future work will be needed to clarify the origin and role of these cells.

Activated cellular pathways in Cluster 14 included complement activation and immune activation, known periportal functions in mice[12,24] (Top DE genes in Cluster 14: *CYP2A7, ENTPD5, CYP3A7, CYP2A6, C4B, EID2, TP53INP2, SULT1A1, ATIC, SERPINH1, SAMD5, GRB14*) (Supplementary Data 1&2). While this cluster is most transcriptionally similar to periportal mouse layers (Supplementary Fig. 8), there was no significant correlation to known layers, likely because the top DE genes (*CYP2A* genes) in this cluster have different roles in mouse and human[25,26] and did not map well with the 94 genes that defined the mouse sinusoidal layers.

Cluster 6 hepatocytes (DE genes: *SEC16B, SLBP, RND3, ABCD3, RHOB, EPB41L4B, GPAT4, SPTBN1, SDC2, PHLDA1, WTAP, ACADM)* have activated cellular pathways in immune cell activation, fibrinolysis, and triglyceride biosynthesis, suggesting a periportal origin. However, since this cluster did not correlate with the mouse sinusoid regions (Supplementary Fig. 8), the origin of this cluster requires further examination.

Our analysis may suggest new aspects of hepatic stem cell biology. The liver displays a high regenerative capacity involving resident stem cells. There is evidence that hepatic stem cells can originate from niches in the canal of Herring, the terminal branch of the intrahepatic biliary system[27]. These stem cells are called hepatic progenitor cells in human and oval cells in rodents. Alpha-fetoprotein (AFP) is described as a marker for hepatic stem cells in human[28]. In our current understanding of the zonated rodent liver, the perivenous compartment is resistant to proliferative signals[29], while the periportal region contains oval cells[30]. Our study challenges this paradigm, since *AFP* is expressed in all clusters of hepatocytes except Cluster 14 (Fig. 4c.v)—we were unable to identify a discrete cluster specific to hepatic progenitor cells. We selected hepatocytes with detectable *AFP* expression, and generated a ranked list of genes DE in *AFP*⁻ vs *AFP*⁺ hepatocytes (Supplementary Data 5), which we used to examine enriched cellular pathways cells in the *AFP*-expressing cells. Enriched pathways in AFP⁺ cells included those for cellular division and IL-6/7 signaling (Supplementary Fig. 10, Supplementary Data 5). This supports the notion that these cells may be hepatic stem cells since IL-6 is a key cytokine for the proliferation of hepatocytes[31]. Our findings raise the possibility of a less localized and more heterogeneous model of hepatic progenitor cells in the human liver.

**Liver endothelial cells**. Our current knowledge of human LSECs is limited to immunofluorescence, flow cytometric studies[32,33], bulk RNA-seq studies on sorted LSECs[34], and functional properties assessed during brief in vitro cultures[35,36]. The liver vascular endothelium, made up of LSECs and the endothelium of blood vessels, provides a dynamic barrier between the blood and the liver microenvironment. LSECs are defined as scavenger endothelia that use clathrin-mediated endocytosis for the clearance of macromolecules from the blood[36]. Recently, immunofluorescent staining was used to describe the zonation of human LSECs and a population of CD36[hi]CD32B[−]CD14[−] LYVE-1[−] LSECs in Zone 1 of the hepatic acinus (the periportal area). A separate LYVE-1[+], CD32B[hi], CD14[+], CD54[+], CD36[mid-lo] LSEC population was located in Zones 2 and 3[32] (central venous). While this work validates the notion of zone-specific heterogeneity within LSECs, it remains limited by the number of tested antigens and availability of validated immunological reagents.

We observed three endothelial cell populations (clusters 11–13; Figs. 2f, 3a, b, f, 5; Supplementary Fig. 11), which were less proliferative than immune cells (Figs. 2e & 5a), and expressed CALCRL (Fig. 5c.i) and RAMP2, suggesting sensitivity to adrenomedullin signaling[37]. The most abundant endothelial cell cluster (Cluster 11) displayed enriched expression of F8, PECAM1, with little expression of CD32B, LYVE-1, STAB2, and CD14 (Fig. 5c, Supplementary Fig. 11). (Top DE genes: MGP, SPARCL1, TM4SF1, CLEC14A, ID1, IGFBP7, ADIRF, CTGF, VWF, CD9, C7, SRPX, ID3, CAV1, GNG11, AQP1, HSPG2, EMP1, SOX18, CLDN5). In line with previous work[32], we propose that these endothelial cells are likely periportal LSECs (Zone 1). The second most abundant endothelial cell population (Cluster 12) was characterized by the enriched expression of CD32B, LYVE1, STAB2, with little expression of VWF (Top DE genes: CCL14, CLEC1B, FCN2, S100A13, FCN3, CRHBP, STAB1, GNG11, IFI27, CLEC4G, CLDN5, CCL23, OIT3, RAMP3, SGK1, DNASE1L3, LIFR, SPARC, ADGRL4, EGFL7, PCAT19, CDKN1C) (Fig. 5c.ii–iii). Based on prior histological examinations of LSEC zonation[32], we propose that this is an LSEC cluster that is central venous in origin (Zone 2/3). The least abundant hepatic endothelial cell population (Cluster 13) was characterized by low or no expression of LSEC markers (LYVE1, STAB2, CD32B). These cells are likely non-LSEC endothelial cells including central vein and portal arterial and venous endothelial cells based on the expression of ENG (protein alias CD105) and PECAM1 (protein alias CD31) as has been described in the human liver via immunohistochemistry[32]. (Top DE genes: RAMP3, INMT, DNASE1L3, LIFR, PTGDS, C7, CTGF, TIMP3, RNASE1, ID3, ENG, MGP, PCAT19, HSPG2, GPM6A, PTPRB, VWF, FAM167B, SRPX, LTC4S, IFI27). To confirm our DE findings at the protein level and to strengthen our assertions regarding LSEC zonation, we examined the expression of CD32B by immunohistochemistry and found that CD32B protein expression was concentrated in the central venous (Zones 2/3) areas of the sinusoid, including the central vein, with limited periportal staining (Fig. 5e.i–vi), confirming previous findings[32]. A pairwise pathway analysis comparing all endothelial cell clusters to each other revealed that clusters 11 and 13, periportal LSECs and portal endothelial cells, respectively, were functionally very similar (Supplementary Fig. 12), with periportal LSECs having significantly enriched pathways in vessel development, cell-cycle arrest, cell junction, and apoptosis. Both periportal LSECs and portal endothelial cells showed enriched pathways in translation, targeting ER, and TNF activation in comparison to central venous LSECs (Fig. 5d, Supplementary Fig. 13). Central venous LSECs (Cluster 12) displayed highly enriched immune pathways, including innate immunity, phagocytosis, leukocyte activation, bacterium defense, in comparison to Clusters 11 and 13 (Fig. 5d, Supplementary

Fig. 13). These results present a rich marker description and functional profile of human liver endothelial cells. However, the characterization of endothelial cell clusters as sinusoidal, rather than arterial or vascular endothelium, requires visualization of fenestrations and specific scavenger receptor functions[38] that are uniquely characteristic of sinusoidal endothelial cells. These descriptions will require optimization of LSEC culture conditions, identification of surface markers that distinguish these populations, and confirmation of protein expression of these markers by methods such as CITE-seq[39], followed by flow sorting and in vitro culture to visualize fenestrations.

**Cholangiocytes**. Cholangiocytes are epithelial cells which line bile ducts that comprise 3–5% of the cells in the human liver and generate 40% of the total bile volume[40]. Intrahepatic cholangiocytes originate from hepatoblasts during embryonic development. Based on the duct diameter size, cholangiocytes are categorized as small and large, each with different secretory functions and sensitivity to injury[20]. The circumference of small bile ducts is formed by 4–5 cholangiocytes and by 10–12 cholangiocytes in larger ducts[41]. Recent studies have outlined single-cell transcriptomic profiles from rodent cholangiocytes[42,43], however the transcriptional profile of human cholangiocytes has not been described due to challenges in isolating these cells[44]. Using our dissociation protocol, we were able to identify a cholangiocyte cell cluster in all five livers profiled (Figs. 2f, 3a, b). Differentially expressed genes in this cluster (Cluster 17) included SOX9, EPCAM, and KRT19 (Fig. 6a, b, f), in accordance with genes that were previously described as enriched in primary human cholangiocytes[45]. Importantly, we found a population of cells which displayed upregulated expression of genes encoding secretory and inflammatory pathways (Top DE genes: TFF2, SCGB3A1, FXYD2, KRT7, DEFB1, CD24, TFF3, LCN2, KRT19, CXCL1, PIGR, TFF1, CXCL6, LGALS2, TACSTD2, ELF3, SPP1, MUC5B, LGALS4). This transcriptional profile of cholangiocytes can serve as a reference point for assessing the physiological nature of human stem cell-derived cholangiocytes.

**Hepatic stellate cells**. Hepatic stellate cells (HSCs) are found in the subendothelial space of the liver sinusoid, known as the space of Disse[46]. HSCs are the main storage site for Vitamin A and are major contributors to tissue fibrosis. Upon activation, human HSCs express α-smooth muscle actin (ACTA2) and begin to lay down extracellular matrix, which is composed of collagen (e.g., COL1A1, COL1A2). Cluster 20 was identified as HSCs based on the expression of ACTA2, COL1A1, TAGLN, COL1A2, COL3A1, SPARC, and the expression of retinol binding protein 1 (RBP1), a vitamin A-associated transcript (Figs. 2f, 7). These genes have been previously described as being upregulated during hepatic stellate cell activation in human liver[46]. Additional genes specific to this cluster include DCN, MYL9 (Fig. 7h–i), TPM2, MEG3, BGN, IGFBP7, IGFBP3, CYR61, OLFML3, IGFBP6, CCL2, COLEC11, CTGF, HGF (Top DE genes). Thus, our data confirms and extends the human HSC transcriptional signature.

**Zonation of human hepatic tissue: the cellular basis for zone-specific gene signatures**. Recently, the zonation of human hepatic tissue was examined by laser capture microdissection and RNA profiling of uninvolved liver tissue, obtained during hepatectomy for metastatic tumors to the liver[24]. In that study, gene expression patterns were described for each zone of the liver lobule. The limitation of such a study is in not knowing the cellular origin of the genes identified: our study is thus complementary and determines the cellular origins of the zonal gene expression signatures. Among the genes previously found to be upregulated in

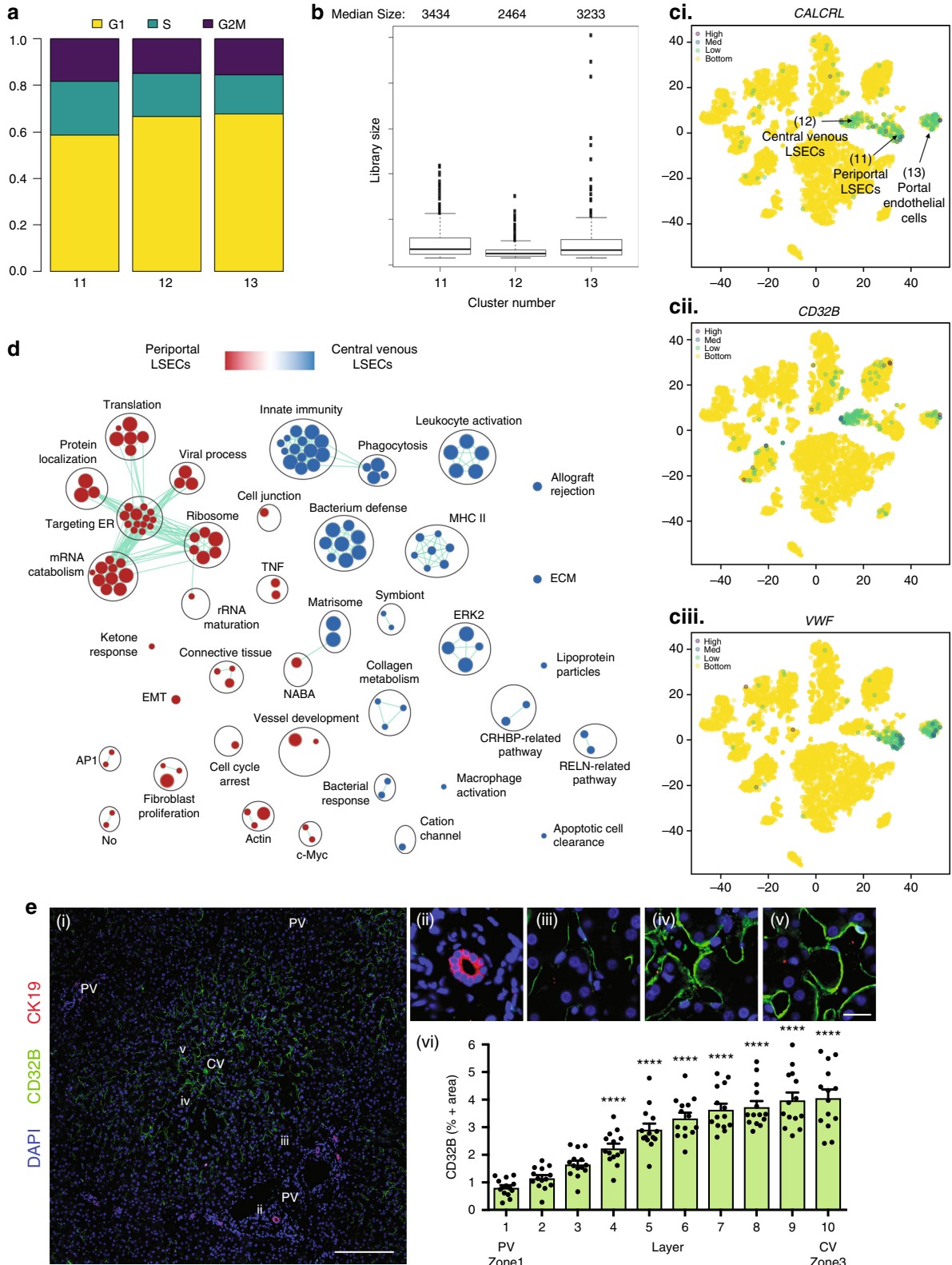

zone 1 (periportal area)[24], we show enriched expression of *CRP, SLPI, LEPR, A2M, CHI3L1, HAL* in Cluster 5 (which correlates with mouse periportal zones); *C7, MGP, ID1, LDB2, CD9, AQP1, SOX18* in portal endothelial cells and zone 1 LSECs (Clusters 11 &13); and *KRT7, KRT19, CXCL6, SFRP5, CLDN10, BICC1, AQP1, ERICH5* in cholangiocytes (known to be enriched in the periportal areas). Among the zone 3 genes previously identified[24],

*RELN* was enriched in zone 2/3 LSECs and *CYP3A4* and *ADIRF* were enriched in Cluster 15 (Supplementary Data 1, 2).

**Intrahepatic immune cells**. Increasing evidence shows that the liver possesses complex immunological properties[1,47–49]. However, the identity, frequency, and phenotype of hepatic immune cells is largely unknown due to the challenges in extracting

**Fig. 5** ScRNA-seq analysis of LSEC populations. **a** Distribution of LSECs by cell-cycle phase (G1, G2/M, S) and LSEC cluster (11, 12, 13). **b** Box plot of library size for each LSEC cluster with median library size (top) and graphically denoted median (dark horizontal line). Outliers (black dots) and interquartile range (black box) are indicated. **c** t-SNE projection of the expression of established LSEC markers in the three identified clusters. **c** (i) CALCRL **c** (ii) CD32B, and **c** (iii) VWF. **d** Pairwise pathway enrichment analysis of genes DE between clusters 11 and 12 defined in Fig 2f. Pathways enriched in periportal LSECs (Cluster 11) are labeled in red and pathways enriched in central venous LSECs (Cluster 12) are indicated in blue. Colored circles (nodes) represent pathways, sized by number of genes they contain. Green lines depict intra- and inter-pathway relationships according to the number of genes shared between each pathway. Black circles group related pathways into themes that are labeled. **e** (i) Immunofluorescence of CD32B distribution in liver zone 1 (portal vein - PV), and 2/3 (central vein - CV). CD32B stains mainly LSEC cluster 12 and CK19 stains periportal ductal cells. **e** (ii, iii, iv, v) are magnified sections corresponding to the indicated roman numerals (white) in (**e** (i)). **e** (i) Scale bar represents 200 μm, **e** (v) scale bar represents 20 μm. Staining was performed on HIER (10 mM Citrate pH 6.0, 95 °C,15 min) treated slides visualized using the matching donkey anti-host antibody and counterstained with DAPI. Slides were scanned and lobules defined as in Supplementary Fig. 10. **e** (vi) Quantification of percent CD32B positive cells in liver zones 1–3. Error bars show the standard error of the mean for at least 10 replicates. Statistical significance evaluated using a one-way analysis of variance (ANOVA) with a Bonferroni post-test ***$P < 0.001$, **$P < 0.01$, *$P < 0.05$. t-SNE: t-distributed stochastic neighbor embedding

immune cells from human liver tissue in a non-biased and viable manner[48]. In mice, a population of KCs has been identified by scRNA-seq[12], but additional immune cell populations were not described. Using our gentle fractionation methods and scRNA-seq analysis, one key goal of our study was to overcome prior challenges and expand the current knowledge of intrahepatic immune cell populations.

**Intrahepatic monocytes/macrophages.** The liver is the solid organ in the body with the largest population of tissue resident macrophages[1,50]. Tissue resident macrophages have been described as the immunological sentinels of the liver[51], but because they are difficult to isolate from humans and have a complex ontogeny[52] they are poorly understood[47]. Traditionally, macrophages are classified as having inflammatory or immunoregulatory properties, as determined by their surface marker phenotype or cellular functions[53]. We previously employed flow cytometry to quantify expression of traditional inflammatory or regulatory macrophage surface markers and found a spectrum of expression in freshly isolated human KCs[13]. Surprisingly, our scRNA-seq analysis consistently revealed the presence of two distinct populations of intrahepatic CD68+ macrophages (Figs. 2f, 3a, b, Supplementary Fig. 14) in all livers studied. CD68+ macrophage population 1 (Cluster 4; Fig. 2f), was characterized by enriched expression of *LYZ, CSTA, CD74*[54], suggesting that this cluster represents inflammatory macrophages (Top DE genes: *S100A8, LYZ, S100A9, HLA-DPB1, S100A12, RP11-1143G9.4, EVI2A, HLA-DPA1, VCAN, S100A6, CXCL8, HLA-DRA, MNDA, TYROBP, HLA-DRB1, FCN1, HLA-DQA1, IL18, C1QC, CD74, HLA-DRB5*).

CD68+ macrophage population 2 (Cluster 10; Fig. 2f) was characterized by enriched expression of *CD5L, MARCO, VSIG4, CPVL, CD163, CCDC88A, C5AR1, LIPA, LILRB5, MAF, CTSB, MS4A7, VMO1, RAB31, SLC31A2, TTYH3, VCAM1, KLF4, HMOX1, AIF1l, TMIGD3* (top DE genes). The DE genes in this cluster suggest that these macrophages have a tolerogenic function. For example, VSIG4 is a co-inhibitory ligand and, in mice, is required to maintain an intrahepatic tolerogenic milieu[55]. Similarly, *HMOX1* (hemoxygenase) knockdown in mice leads to hepatic inflammation[56]. Confirming the unique functionality of the KC populations, pathway analysis revealed that KCs in cluster 10 were enriched for pathways related to tolerance while KCs in cluster 4 were enriched for inflammatory pathways (Fig. 8c).

A key finding of our study is the presence of two distinct populations of human liver macrophages, seeming to segregate into pro-inflammatory and immunoregulatory phenotypes. Our gene list identifies several markers that are unique to one or the other population (for example, *MARCO* (MAcrophage Receptor with COllagenous structure) is only expressed in non-inflammatory KCs), a finding that can be exploited in tissue-

based studies. Using flow cytometry, we observed a subpopulation of macrophages which expressed MARCO on the cell surface (Fig. 8d, Supplementary Fig. 15). By immunohistochemistry staining for MARCO, MARCO+ cells are concentrated in the periportal areas (Fig. 8e, f; Supplementary Figs. 16, 17). We then examined the human macrophage expression profiles that characterize both CD68+ populations in the context of the mouse macrophage ontogeny literature. Human *CD68+ MARCO*+ cells appear to be transcriptionally similar to long-lived, sessile, liver resident KCs identified in mouse[57–59], with these cells expressing high levels of *VCAM1*[60], *CD5L*[60], *HMOX1, MRC1*[61], *CD163*[60], *M4SA7*, and *VSIG4*[58,60]. *CD68+ MARCO*− macrophages have a similar transcriptional profile to inflammatory, recently recruited macrophages including decreased expression of *CD163*[58] and increased expression of *PLBD1*[57]. To further examine this point using functional assays, we stimulated both populations of macrophages in vitro and examined cytokine secretion via intracellular cytokine staining. We found that MARCO-positive macrophages secreted less TNF-α in response to LPS/ IFN-γ stimulation than MARCO-negative CD68+ macrophages, suggesting that CD68+MARCO- cells are more pro-inflammatory (Fig. 8d, Supplementary Fig. 15). The expression of MARCO in the tumor microenvironment has been linked to poorer outcomes in human breast cancer[62]. MARCO has also been examined in preclinical mouse colon cancer models with the observation that MARCO expression defined a subtype of suppressive tumor-associated macrophages (TAMs). These TAMs could be polarized to an inflammatory phenotype by anti-MARCO antibody which promoted tumor immunogenicity[63]. These findings provide a point of reference for examining the role of intrahepatic monocyte/macrophage subsets in the establishment and progression of liver disease.

**Liver resident T cells.** The human intrahepatic T cell phenotype has been examined via flow cytometry evaluations of enzymatically and mechanically dispersed biopsies taken after reperfusion of donor organs during transplantation[64]. In this study, the frequency and phenotype of intrahepatic T cells differed from that of peripheral circulating T cells. In our study, we expand on these findings by examining the gene expression patterns that characterize T cell populations in the flushed human liver using cells obtained from a segment of liver tissue obtained prior to reperfusion.

The T cell repertoire within the human liver can be categorized into two broad groups: conventional and unconventional. Conventional T cells consist of CD8+ and CD4+ cells expressing αβ-chain TCRs that recognize antigen in the context of MHC class I and class II molecules respectively. Conventional T cells comprise up to 86% of all CD3+ T cells within the human liver[64]. Unconventional T cells can express either αβ or γδ TCRs but are

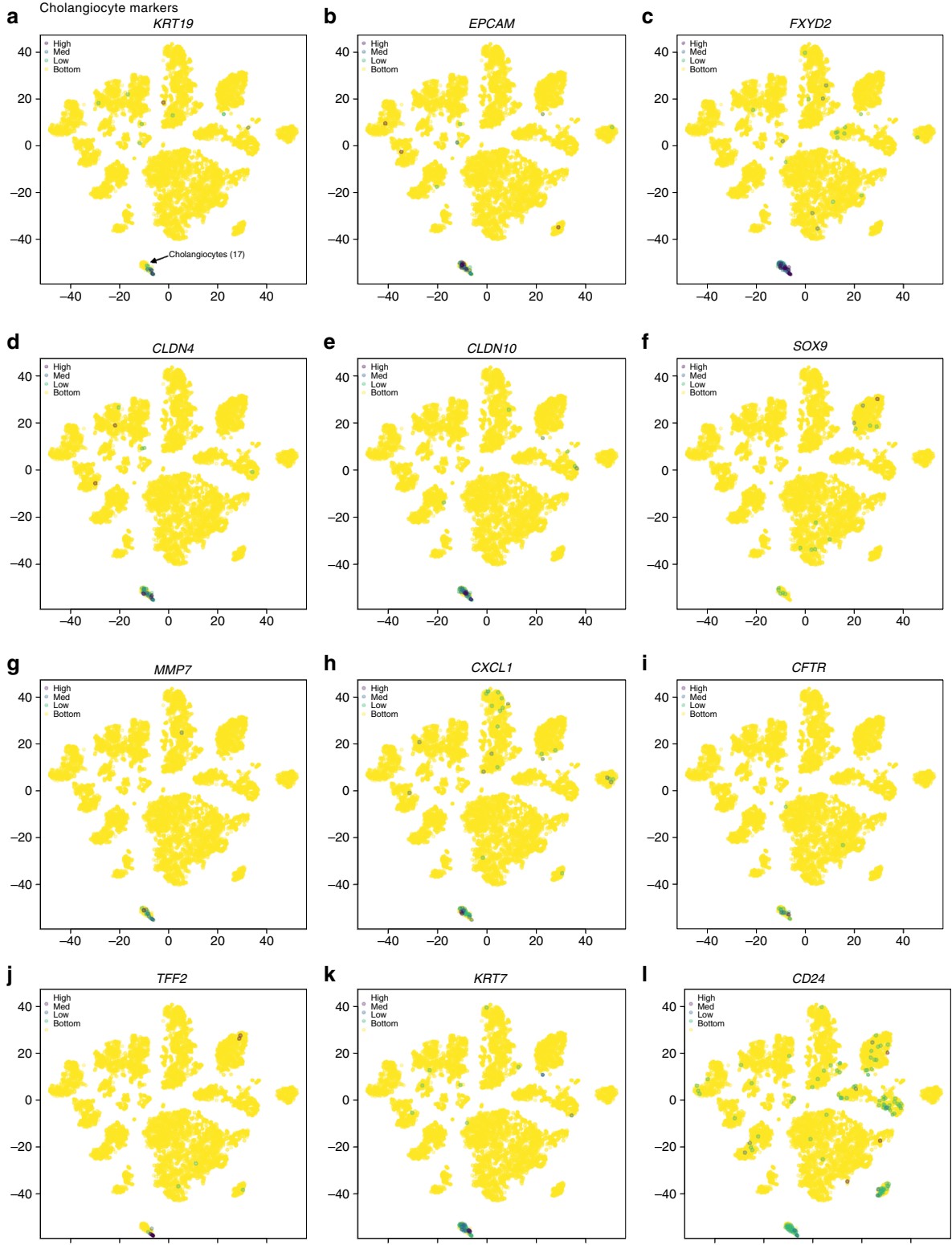

**Fig. 6** ScRNA-seq analysis of the cholangiocyte population. *t*-SNE plots showing relative distribution of commonly expressed cholangiocyte genes in the healthy NDD liver. Protein alias described in parentheses if different from gene name. Expression of **a** *KRT19* (CK19), **b** *EPCAM*, **c** *FXDY2*, **d** *CLDN4*, **e** *CLDN10*, **f** *SOX9*, **g** *MMP7*, **h** *CXCL1*, **i** *CFTR*, **j** *TFF2*, **k** *KRT7* (CK7), **l** *CD24*. Legend for relative expression of each marker from lowest expression (yellow dots) to highest expression (purple dots) (top left). *t*-SNE: t-distributed stochastic neighbor embedding

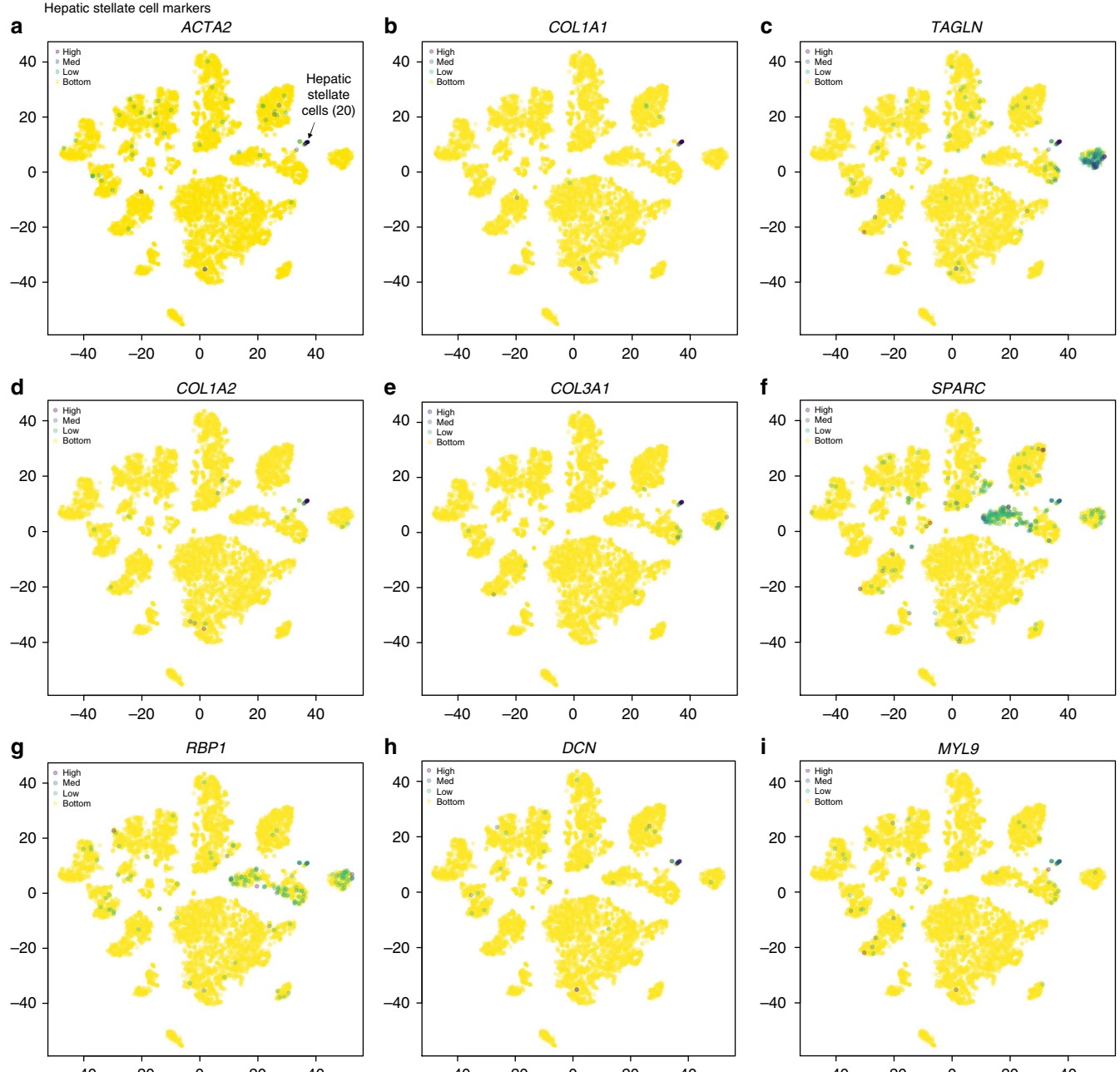

**Fig. 7** ScRNA-seq analysis of hepatic stellate cell population. *t*-SNE plots showing the relative distribution of commonly expressed hepatic stellate cell genes in the healthy liver. Protein alias described in parentheses if different from gene name. Expression of **a** *ACTA2*, **b** *COL1A1*, **c** *TAGLN*, **d** *COL1A2*, **e** *COL3A1*, **f** *SPARC*, **g** *RBP1*, **h** *DCN*, and **i** *MYL9*. Legend for relative expression of each marker from lowest expression (yellow dots) to highest expression (purple dots) (top left). *t*-SNE: t-distributed stochastic neighbor embedding

not restricted by MHC and do not recognize classical peptide antigens[65]. In mice, the microbial antigens that enter the liver by the portal vein following digestion have been found to sustain γδ T cell homeostasis and activation[66]. The relative abundance of subsets of unconventional T cells within the human liver remains to be defined.

In our five healthy NDD livers, we identified three clusters of CD3+ T cells. The most abundant population of T cells (Cluster 2) was characterized by the expression of *CD2, CD3D, TRAC, GZMK, CCL5, CCL4L2, PYHIN1, TRBC1, TRBC2, GZMA, CD3E, JUNB, CD69, IL7R, DUSP2, IFNG, LTB, IL32, CD52* (Top DE genes) (Fig. 9a, Supplementary Data 1, 2). These appear to be αβ T cells due to their expression of CD3 and αβ-chain TCRs.

Within this population, cells show enriched expression of *CD69* and *CD8A* (Supplementary Data 1, 2). CD69 is a marker for recently activated T cells, and is enriched in tissue resident memory T cells when compared to circulating memory T cells[67,68]. However, multiple studies have shown that CD69+ CD45RA− tissue resident memory T cells do not express activation markers such as CD25 (*IL2RA*) or CD38, indicating that the predominant cell in this cluster likely represents tissue resident memory T cells instead of newly activated T cells[67]. Furthermore, CD69 antagonizes the function and downregulates S1PR1, which is needed for T cell egress from tissues[47,48]. Since we observed no enrichment of expression of *S1PR1*, *CD38*, or *IL2RA*, all genes characteristic of newly activated non-memory

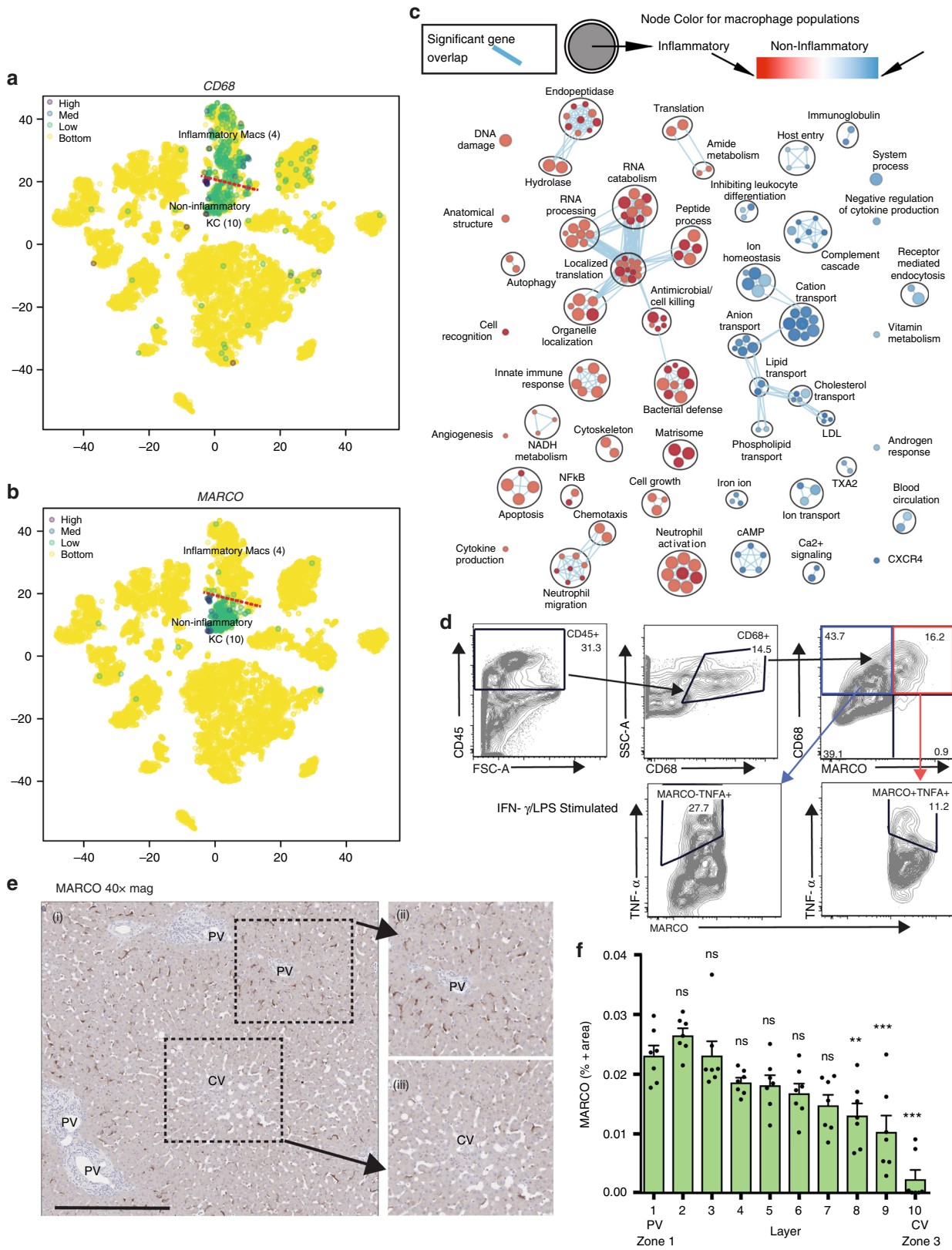

T cells, this further supports our identification of these cells as tissue resident memory T cells. The second cluster of CD3+ cells (Cluster 9), was a population of *CD3D* expressing cells with enriched expression of *TBX21* (aka T-bet), *KLRB1* (protein alias CD161), *FCGR3A* (CD16), *NKG7*, and *GNLY* (NKG5) with enriched expression of the TCR delta chain *TRDC* and the TCR

gamma chain *TRGC1*. The expression profile of these cells suggests that the predominant population in this cluster may be unconventional γδ T cells[65] (top DE genes: *GNLY, PTGDS, GZMB, S100B, FGFBP2, NKG7, PRF1, KLRF1, HOPX, CST7, KLRD1, CTSW, SPON2, IFITM1, GZMA, CD247, CLIC3, CD7, ADGRG1, CCL5, TRDC*). A third cluster of T cells (Cluster 18)

**Fig. 8** ScRNA-seq identifies two distinct populations of human liver resident macrophages/monocytes. **a, b** t-SNE projection of 8444 liver cells, with each cell colored based on expression of **a** CD68 and **b** MARCO. **c** Pairwise pathway enrichment analysis comparing gene expression in the two *CD68*[+] macrophage clusters defined in Fig 2f. Pathways enriched in non-inflammatory KCs are labeled in blue and pathways enriched in inflammatory KCs are indicated in red. Colored circles (nodes) represent pathways, sized by number of genes they contain. Green lines depict intra- and inter-pathway relationships according to the number of genes shared between each pathway. Black circles group related pathways into themes that are labeled. **d** Flow cytometry data showing the response of monocytes/macrophages in total liver homogenate cell suspensions to stimulation with 1 μg/ml LPS and 25 ng/ml IFN-γ. Cells were stained with anti-human CD45 (clone: HI30), anti-CD68 (clone: Y1/82 A), anti-MARCO (polyclonal; Invitrogen, PA5-26888, goat anti-rabbit secondary antibody), and anti-TNF-α antibodies. Full gating strategy and controls shown in Supplementary Fig. 15. **e, f** Distribution of MARCO-positive cells in liver zones 1–3. Scale bar represents 500 μm. Staining was performed on 5–7 μM slices cut from formalin-fixed, paraffin-embedded resected liver tissue. Using anti-MARCO (clone: Invitrogen, PA5-26888) and anti-CD68 (clone: PG-M1) at ×40 magnification. **f** Quantification of percent MARCO-positive cells in liver zones 1 to 3. Error bars show the standard error of the mean for at least seven replicates. Statistical significance evaluated using a one-way analysis of variance (ANOVA) with a Bonferroni post-test ***$P < 0.001$, **$P < 0.01$, *$P < 0.05$

was identified as having enriched expression of *CD3*, little *CD4* and *CD8* expression with enriched expression of *TRDC* (Supplementary Data 1&2). The most highly DE gene in this cluster was *STMN1*, which is expressed on proliferating cells of various lineages[69]. Cell-cycle analysis revealed that 79% of the cells in this cluster were in G2M (Fig. 2e, Supplementary Fig. 18). *MKI67* was also upregulated as a further indication that the cells in this cluster are dividing. Other highly expressed genes in this cluster were *GNLY, NKG2A, TYMS*, and *TOP2A*, suggesting that these cells might be phosphoantigen-reactive γδ T cells due to shared enriched genes previously identified in phosphoantigen-reactive γδ T cells isolated from human PBMCs[70] (top DE genes *STMN1, HMGB2, TYMS, KIAA0101, MKI67, UBE2C, TUBA1B, TRDC, ASPM, CENPA, TOP2A, GNLY, PCNA, AURKB, BIRC5, NUSAP1, TROAP, TUBB, H2AFX, CENPF, CCNB1, H2AFZ*).

**NK-like cells**. Recently, three populations of innate-like lymphocytes, ILC1, ILC2, and ILC3 were described[71]. ILC1 cells include NK cells and other ILCs characterized by expression of T-bet and production of IFN-γ. ILC 2 include cells that express GATA3, BCL11B, and GFI1 and produce IL-4, IL-5, IL-13, and amphiregulin (AREG). ILC3 are thought to include lymphoid tissue inducer (LTi) cells and cells that are positive or negative for the natural cytotoxicity receptors (NCRs) NKp44; they are defined by the expression of RORγt and the production of IL-17A, IL-17F, and IL-22. In studies of human liver cells obtained by flow-based cell sorting of enzymatically and mechanically dissociated hepatic resection tissue, perforin, and granzyme staining revealed that human liver NK cells rapidly respond to antigens and malignant cells by quickly releasing lytic granules[36]. By flow cytometry, CD16− NK cells make up to 50% of NK cells in the human liver[36]. We found a population of NK-like cells (Cluster 8) characterized by enriched expression of *CD7*, C–type lectin receptor *KLRD1* (CD94), *GZMK* (Granzyme K), *NCR1* (NKp46), and *NCAM1* (CD56), without upregulated expression of *FCGR3A* (CD16) or *ITGA1* (CD49a) (Supplementary Data 1&2). This population also showed enriched expression of *EOMES*, suggesting that the predominant population in this cluster may correlate with the long-lived NK cell population previously described in human liver biopsies by flow cytometry[72]. The top DE genes of Cluster 8 are: *CD7, CMC1, XCL2, KLRB1, XCL1, KLRC1, KLRF1, IL2RB, CD160, CCL3, KLRD1, NKG7, TXK, ALOX5AP, TRDC, CD69, TMIGD2, CLIC3, GZMK, DUSP2, MATK, IFITM1, CCL4, CD247*. Along with conventional T cells, γδ T cells, and NK-like cells, it is known that NKT cells, MAIT cells and atypical T cells populations are found in the liver[73,74]. In our initial analysis, we identified clusters that corresponded to T and NK-like cells, but our data does not currently have the resolution to characterize the less-frequent immune cell populations from the TLH analysis and we suggest that these clusters are likely multiple cell populations. MAIT cells in

particular are a subset of T cells with an αβ TCR characterized by a semi-invariant TCR alpha (TCRα) chain and CD161 expression. These cells likely fall within Cluster 2, for example CD161 (*KLRB1*) is expressed on 52% of cells in Cluster 2 (Supplementary Data 1, 2). In the future, we will employ TCR clonotyping to examine the transcriptional signature of CD161+ αβ TCR+ cells expressing the known semi-invariant TCRs characteristic of MAIT cells (*TRAV1-2/TRAJ12/20/3*). As well, in cluster 8 (NK-like), 23% of the cells express NKp46 (*NCR1*) and 18% express CD56 (*NCAM1*)(Supplementary Data 1&2). This suggests that additional cell populations are clustering with CD56+ NKp46+ cells due to similarities in gene expression. To address this issue, we focused on the T and NK cell populations and carried out a sub-analysis comparing clusters 2, 8, 9, and 18 to one another. This analysis yielded nine populations of conventional and unconventional T cells and NK-like cells that comprise cells with inflammatory or immunoregulatory properties (Supplementary Data 2, 6 and Supplementary Fig. 19). These results provide a description of baseline T and NK cell transcriptomes in the human liver. Further characterization of these cell populations will require a more cells and additional single-cell analysis tools, including surface protein labeling (CITE-seq)[39] and simultaneous characterization of the T cell receptor clonotype using single-cell V(d)J sequencing [https://www.10xgenomics.com/solutions/vdj/]. These approaches will improve our ability to detect and identify these important cell populations in future analyses.

**B cells**. The frequency and role of hepatic B cells remains a topic of debate[48]. Flow cytometric analyses of human liver tissue indicated that CD19+ B cells account for up to 8% of all lymphocytes[64], while B cells have been found to make up as much as 37.5% of the mouse intrahepatic immune cell population[75]. Information on the phenotype and function of these B cells is limited due to their small number and the difficulty in isolating and analyzing hepatic B cells[48]. Recently, IgA-producing plasma cells of GALT (gut-associated lymphoid tissue) origin were found to be enriched in human liver periportal areas by histology[76]. Using scRNA-seq we were able to identify distinct populations of B cells with different stages of development. We found two populations of liver resident B cells which appeared to be plasma cells and antigen inexperienced B cells (Figs. 2f, 3a, b, 9d i–iv, e i–iv). Cluster 16 identified a subset of cells that were enriched for the expression of *IGHD, CD19, MS4A1* (protein alias CD20), *CD22, CD52* without expression of either *CD27* or *CD138* which suggest that this cluster is comprised of mature antigen inexperienced B cells[77,78] (top DE genes: *MS4A1, LTB, CD37, CD79B, CD52, HLA-DQB1, TNFRSF13C, TCL1A, LINC00926, STAG3, IGHD, BANK1, IRF8, BIRC3, P2RX5, RP11-693J15.5, RP5-887A10.1, VPREB3, CD22, CD74, SELL*).

Plasma cells are terminally differentiated B cells that reside in tissue and continuously synthesize and secrete antibodies directed

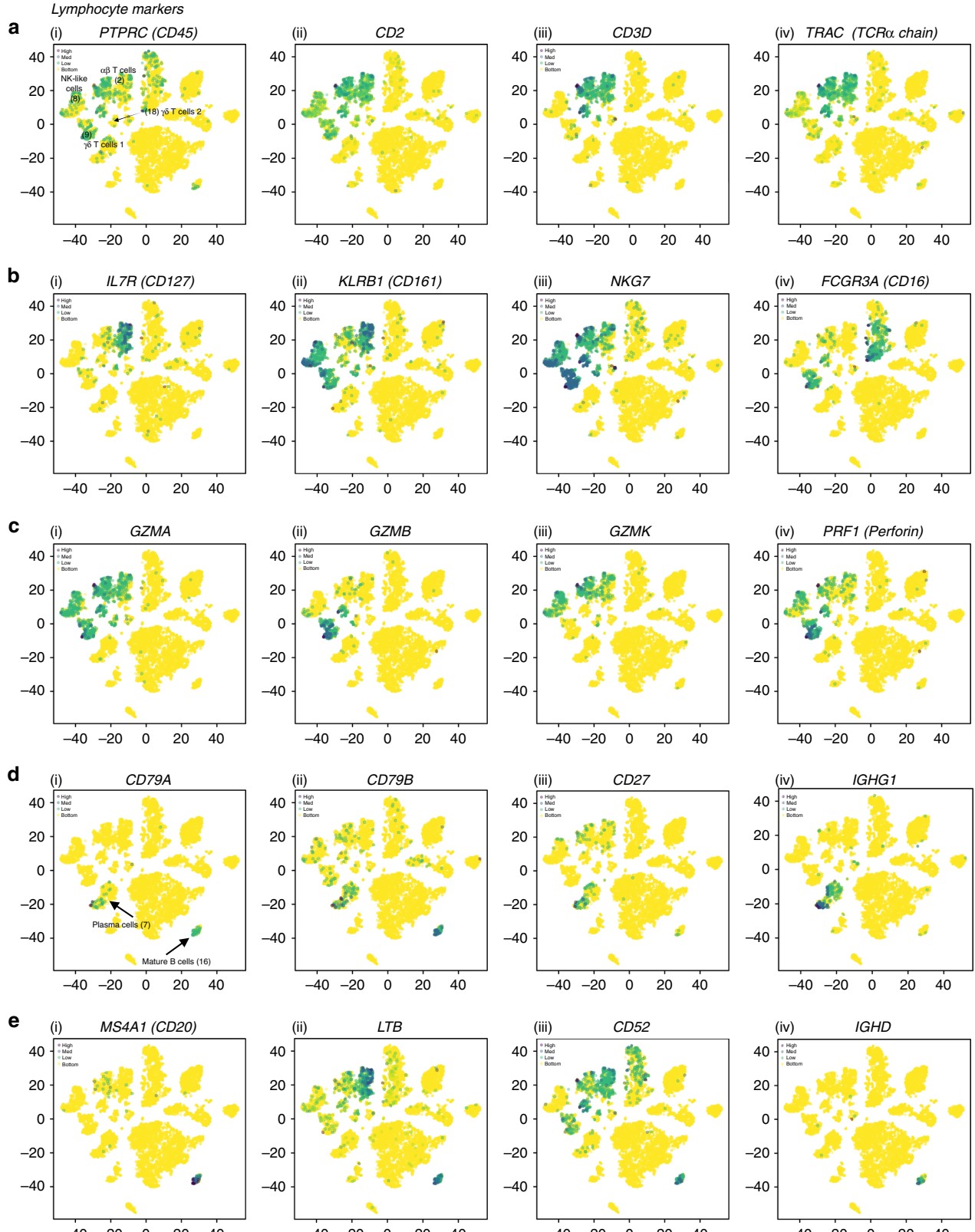

**Fig. 9** The distribution of commonly expressed lymphocyte genes in the healthy liver. **a–c** i–iv *t*-SNE plots showing the relative distribution of common αβ T cell, γ∂ T cell, and NK cell markers. **d** i–iv Common markers of antibody-secreting B cells (plasma cells). **e** i–iv Common markers of mature B cells. Legend for relative expression of each marker from lowest expression (yellow dots) to highest expression (purple dots) (top left)

against specific antigens[79]. The cells of Cluster 7 were enriched for the expression of immunoglobulin heavy and light chains, *CD27*, *CD38*[78], and had reduced expression of *MS4A1*(CD20)[80], suggesting that this cluster is comprised of plasma cells (top DE genes: *IGLC2, IGHG1, IGKC, IGHG2, IGHG3, IGHGP, IGLC3, JCHAIN, IGHA1, IGHG4, IGHA2, IGHM, IGLV3-1, IGLC7, MZB1, CD79A, SSR4, IL16*). Our baseline description of B cell transcriptional profiles in the human liver will be useful in any examination of liver diseases modulated by B cells, such as alcohol-induced hepatitis[76].

## Discussion

A key problem hindering the study of the human liver is difficulty in accessing fresh tissue. Single-cell genomics-based studies of liver tissue can maximize the unbiased information we can extract from these limited samples. Previous approaches to understanding the cellular makeup of the liver via tissue-wide genomics studies have clear limitations. In particular, they can obscure the critical contributions of individual cell populations. Our single-cell description of tissue resident hepatic cells (Figs. 2f & 10, Supplementary Figs. 20–24) adds to our understanding of the cellular landscape of human organs[6,9], and provides a map of the baseline liver cellular network.

Our human liver single-cell transcriptional profile includes more discrete cellular populations than did a previous scRNA-seq examination of the mouse liver[12]. In the mouse study, scRNA-seq and smFISH were employed to spatially reconstruct the mouse liver central vein to portal node based on expression levels of landmark genes. The authors were able to identify three clusters of hepatic cells (hepatocytes, endothelial cells, and KCs). They found that 50% of the hepatocyte genes displayed restricted expression based on their zonation profiles. Importantly, they suggested a key role for the mid lobule in the expression of bile acid synthesis genes. Their paper challenges the classical classification of liver zonation of periportal vs. central venous by identifying the mid-lobule layers as key players in metabolism. The mouse KC population was predominantly pro-inflammatory, expressing *IRF7*, *SPIC*, and *CLEC4F*, while endothelial cells were also pro-inflammatory, expressing *IL1A*. We attribute the differences in the number of cell populations detected in our study to biological differences between livers from laboratory mice and humans, differences in dissociation methods, and the fact that we did not employ sorting to remove dead or low viability cells.

Our data is the first description of distinct macrophage populations in the human liver with unique functional pathways identified. How specific macrophage populations contribute to liver regeneration and the development of liver disease is a topic of ongoing discussion and it has been shown that macrophage subpopulations predominate during liver diseases (i.e., transplant liver graft rejection and liver cancer)[81]. Thus, the characterization of these macrophage populations in human provides a valuable framework for examining the role of macrophage subpopulations in liver disease.

An important conclusion from our work is that liver tissue preparation methodology and bulk liver homogenate viability has a significant impact on the ability to transcriptionally profile hepatic cell populations. Hepatocyte populations are particularly susceptible to dissociation effects. Furthermore, due to the heterogeneity of morphologies of the cells that comprise the liver, it is reasonable to suspect that not all cell types are captured with equal efficiency. Thus, our map identifies populations, but not necessarily their actual frequency within the original liver tissue. This is a caveat that should be considered in any interpretation of scRNA-seq results.

Confounding factors to be considered when interpreting the data in this study are that while the caudate lobes obtained were from clinically acceptable, healthy liver grafts, these NDD liver grafts are mildly inflamed[23]. Furthermore, this study was limited by viable cell numbers and as such, deeper evaluations may uncover additional populations of intrahepatic immune cells. While our comparison of human to murine hepatocyte transcriptional profiles supports a correlation between individual hepatocyte clusters and sinusoidal zonation, the origin and identity of human hepatocytes will require additional examination. Future scRNA-seq/immunohistochemistry studies with larger numbers of samples that have a smaller variability in pre-analysis viability will be better able to definitively comment on the frequency of these populations, and their sinusoidal zonation. We will complement our approach with simultaneous measurement of cell-surface protein expression by CITE-seq[39] and assessment of the T cell clonotype[82]. We intend to fully examine the impact of the dissociation protocol on the transcriptomic faithfulness of single-cell profiles by comparing scRNA-seq to single nucleus RNA-seq (sNuc-seq)[83] from frozen tissue samples. This will enable us to avoid proteolytic treatment and minimize gene expression changes resulting from dissociation procedures and may improve our ability to profile particularly dissociation–sensitive cell types.

Taken together, our transcriptional map of the human liver microenvironment provides a framework for understanding the cellular basis of human liver function and disease and provides a benchmark for the development new cell-based and immuno-modulatory therapies to treat and prevent liver disease.

## Methods

**Human liver tissue dissociation**. Human liver tissue was obtained from livers procured from deceased donors deemed acceptable for liver transplantation. Samples were collected with appropriate institutional ethics approval from the University Health Network (REB# 14-7425-AE) and processed as described previously[13,14] and detailed below. In all cases, patient demographics were collected and stored securely in an anonymized fashion.

During organ retrieval, donor liver grafts were perfused in situ with cold (HTK) solution (Methapharm) to thoroughly flush circulating cells, leaving only tissue resident cells that are then used to prepare a single-cell suspension for scRNA-seq analysis. At our institute, the caudate lobe (segment 1) of the liver is often removed in preparation of the organ for implantation. The removed caudate lobe was then dissociated using the protocol as fully described in [https://doi.org/10.17504/protocols.io.m9sc96e] and detailed below.

Briefly, the caudate lobe was cannulated with two or three 1.2–2 mm diameter irrigation cannulae with olive tips, which were inserted into exposed vessels in the cut surface if the liver lobe (cannulae manufacturer: Ernst Kratz GmbH cat no: 1464LL, 1465LL). The lobe was flushed with HBS + EGTA at 4 °C (perfusion rate of 10 mL/min/cannulae), which removed any residual non-liver resident cells. Single-cell isolation from the resected caudate liver lobe was performed with a modified two-step collagenase procedure[13] (Fig. 1b). Collagenase perfusion was carried out for 15–30 min to limit cellular activation resulting from prolonged enzymatic digestion[15]. This step takes advantage of the Glisson's capsule, the liver's enveloping sheath of connective tissue, which holds the cells together during the collagenase step. Following collagenase treatment, the capsule was cut and the dissociated single cells were collected. All solutions were oxygenated with 95% $O_2$ and 5% $CO_2$ prior to perfusion to limit cellular stress. A red cell lysis step was not employed as the caudate lobes were flushed prior to dissociation and red blood cells were not evident in the cell pellets. A small fraction (1.1%, cluster 19) of the cells profiled showed the expression of *HBB* and erythrocyte specific genes *SLC25A37, CA1, ALAS2* confirming that our retrieval and isolation protocols result in the elimination of the majority of circulating blood cells (Fig. 2f, Supplementary Data 1, 2). We have modified conventional liver perfusion protocols to maintain the tissue at 4 °C for all steps apart from the collagenase perfusion step, which was carried out at 37 °C. Eight to twelve million viable cells were recovered per gram of tissue as determined by Beckman Vi-Cell trypan blue exclusion. Viability for TLHs by trypan blue exclusion varied between 50–90% viable for the five livers profiled. Caudate lobes weighed on average 30.2 g + /−SD of 8.367 g (Supplementary Table 1). TLHs were taken directly after dissociation for scRNA-seq. In certain cases, scRNA-seq was employed to compare TLH to an NPC-enriched fraction (enrichment performed by a 50× *g* centrifugation followed by two wash steps).

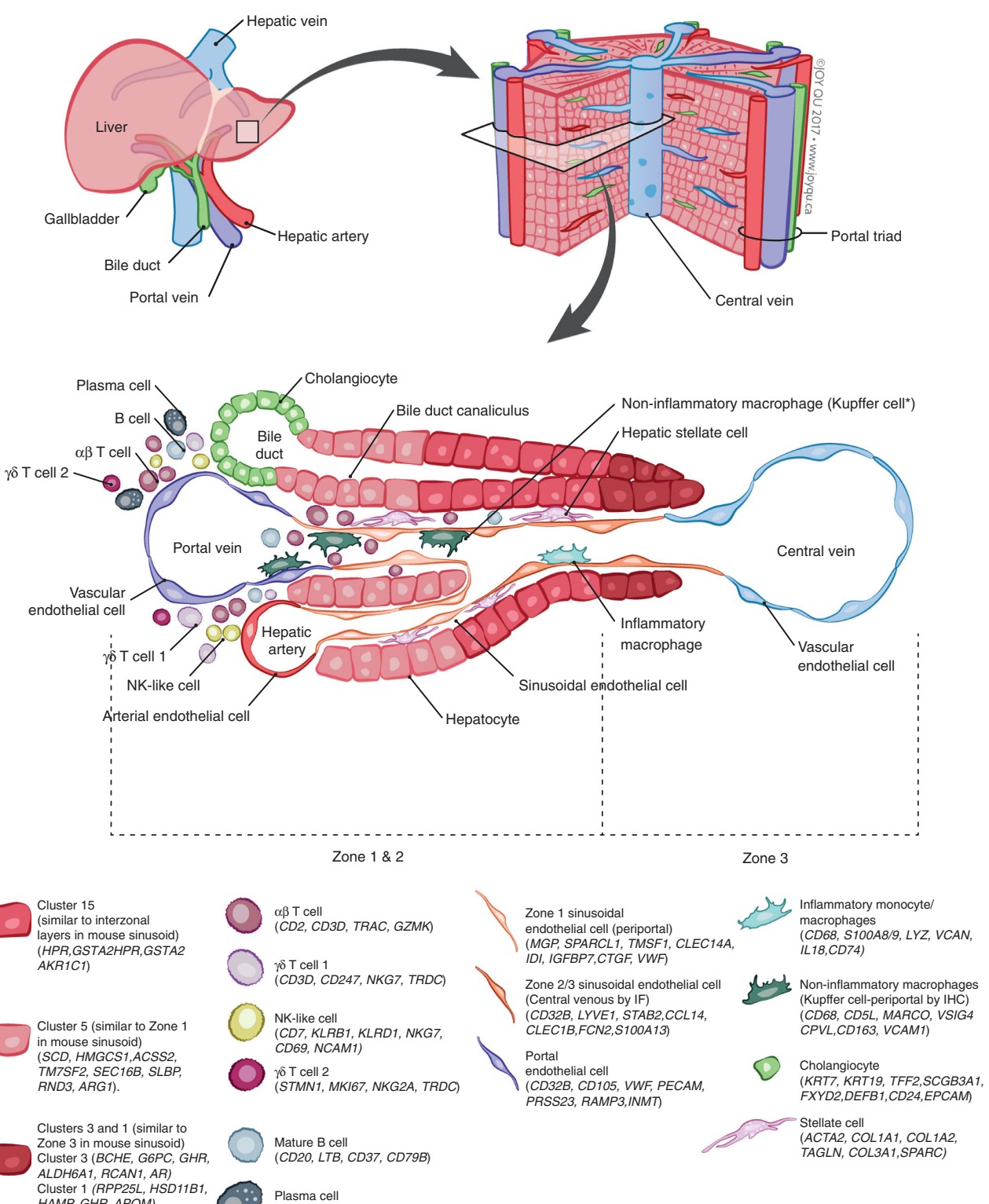

**Fig. 10** Summary map of the human liver. The main "building block" of the liver is the hepatic lobule, which includes a portal triad, hepatocytes aligned between a capillary network, and a central vein. The portal triad is made up of the hepatic artery, the portal vein and the bile duct. Found between the liver sinusoids are parenchymal cells (hepatocytes) and non-parenchymal cells (endothelial cells, cholangiocytes, macrophages, hepatic stellate cells, and liver infiltrating lymphocytes- including B cells, αβ and γδ, T cells, and NK cells). Non-inflammatory macrophages are labeled *Kupffer cells based on their transcriptional similarity to mouse KC. The location of B cells, plasma cells, T cells, and NK cells has yet to be confirmed by immunohistochemical staining of these populations in situ so their location in this schematic is not representative of their zonated distribution. The zonation of hepatocytes was not confirmed by immunohistochemical staining and is inferred as a result of pathway analysis and transcriptional similarity to the zonated gene expression patterns previously shown in mice (Halpern et al.[12])

**10x sample processing and cDNA library preparation**. Samples were prepared as outlined by the 10x Genomics Single Cell 3′ v2 Reagent Kit user guide. Briefly, the samples were washed twice in PBS (Life Technologies) + 0.04% BSA (Sigma) and re-suspended in PBS + 0.04% BSA. Sample viability was assessed via Trypan Blue (Thermo Fisher) and using a haemocytometer (Thermo Fisher). Following counting, the appropriate volume for each sample was calculated for a target capture of 6000 cells. Samples below the required cell concentration as defined by the user guide (i.e., <400 cells/μl) were pelleted and re-suspended in a reduced volume and counted again using a haemocytometer prior to loading onto the 10x Genomics single-cell-A chip. After droplet generation, samples were transferred onto a pre-chilled 96-well plate (Eppendorf), heat-sealed and reverse transcription was performed using a Veriti 96-well thermal cycler (Thermo Fisher). After the reverse transcription, cDNA was recovered using Recovery Agent provided by 10x followed by a Silane DynaBead clean-up (Thermo Fisher) as outlined in the user guide. Purified cDNA was amplified for 12 cycles before being cleaned up using SPRIselect beads (Beckman). Samples were diluted 4:1 (elution buffer (Qiagen): cDNA) and run on a Bioanalyzer (Agilent Technologies) to determine cDNA concentration. cDNA libraries were prepared as outlined by the Single Cell 3′ Reagent Kits v2 user guide with appropriate modifications to the PCR cycles based on the calculated cDNA concentration (as recommended by 10X Genomics).

**Sequencing**. The molarity of each library was calculated based on library size as measured using a bioanalyzer (Agilent Technologies) and qPCR amplification data (Kappa/Roche). Samples were pooled and normalized to 10 nM, then diluted to 2 nM using elution buffer (Qiagen) with 0.1% Tween20 (Sigma). Each 2 nM pool was denatured using 0.1 N NaOH at equal volumes for 5 min at room temperature. Library pools were further diluted to 20 pM using HT-1 (Illumina) before being diluted to a final loading concentration of 14 pM. 150 μl from the 14 pM pool was loaded into each well of an 8-well strip tube and loaded onto a cBot (Illumina) for cluster generation. Samples were sequenced on a HiSeq 2500 with the following run parameters: Read 1—26 cycles, read 2—98 cycles, index 1 —8 cycles. A median sequencing depth of 60,000 reads/cell was targeted for each sample. 10x Genomics web summaries for each liver profiled are found in Supplementary Fig. 25–29.

**Cell clustering, differential expression, and pathway analysis**. Raw sequencing data (bcl files) were converted to fastq files with Illumina bcl2fastq, version 2.19.1 and aligned to the human genome reference sequence [http://cf.10xgenomics.com/supp/cell-exp/refdata-cellranger-GRCh38-1.2.0.tar.gz]. The CellRanger (10X Genomics) analysis pipeline was used to generate a digital gene expression matrix from this data. The raw digital gene expression matrix (UMI counts per gene per cell) was filtered, normalized, and clustered using R [https://www.R-project.org/]. Cell and gene filtering was performed as follows: Cells with a very small library size (<1500) and a very high (>0.5) mitochondrial genome transcript ratio were removed. Genes detected (UMI count > 0) in less than three cells were removed.

Normalization was performed in the scran R package using the default implementation of their pool and deconvolute normalization algorithm[84,85]. Briefly, this normalization method proceeds as follows: hierarchical clustering using a distance metric derived from Spearman's correlation is performed to subset the data into more homogenous groups. Within each group, cell-wise scaling factors are determined, and then normalization is performed between groups. Scaling factors per cell were determined by pooling random subsets of cells, summing their library sizes, and comparing to average library size across all cells in the group. This is iteratively performed, and the cell-wise scaling factors can be deconvolved from the set of pool-wise scaling factors. This method is robust to the sparsity of the data and respects the assumption of minimal differential gene expression common to most normalization methods.

After normalization, clustering is performed using standard Seurat package procedures[86]. Briefly, principal component analysis was used to reduce the number of dimensions representing each cell. The number of components used was determined based on the elbow of a scree plot. A shared nearest neighbor graph was built from distances computed in principal component space. A smart local moving algorithm was used to identify communities in the graph. Selection of a biologically relevant number of clusters was based on differential expression between neighboring clusters. Differential expression between clusters was calculated using a likelihood-ratio test for single-cell gene expression implemented in Seurat at a family-wise error rate of 5%. Neighboring clusters in principal component space were identified as the next-nearest cluster to each cell after the cell's assigned cluster. Clusters were visualized using t-distributed Stochastic Neighbor Embedding of the principal components (spectral t-SNE) as implemented in Seurat. Cell-cycle phases were predicted using a function included in Seurat that scores each cell based on expression of canonical marker genes for S and G2/M phases (Fig. 2e, Supplementary Data 7). The cell-type identities for each cluster were determined manually using a compiled panel of available known hepatocyte/immune cell transcripts. Pathways enriched in specific clusters in Figs. 5d, 8c and Supplementary Figs. 10, 12, 13 were elucidated via a pairwise analysis using the Gene Set Enrichment Analysis (GSEA) software from the Broad Institute (software.broadinstitute.org/GSEA)(version 3.0). Pathway enrichment analysis examining active cellular pathways in hepatocyte clusters in Fig. 4d was performed using

Gene Set Variation Analysis (GSVA)[87] software from Bioconductor (version 1.28). Human_GOBP_AllPathways_no_GO_iea_November_01_2017_symbol.gmt from [http://baderlab.org/GeneSets] was used to identify enriched cellular pathways in GSVA and GSEA analysis. Highly related pathways were grouped into a theme and labeled by AutoAnnotate (version 1.2) in Cytoscape (Version 3.6.1). GSEA and GSVA results were visualized using the Enrichment Map app[88] (Version 3.1) in Cytoscape (Version 3.6.1). R scripts for the data processes and all GSEA raw data (Figs. 5d, 8c and Supplementary Figs. 10, 12, 13 found at: [https://github.com/BaderLab/singleLiverCells].

**Human/mouse correlation analysis**. Human-mouse one-to-one orthologous genes were identified from the Ensembl database[89]. Using the significantly ($p < 1 \times 10^{-25}$) and DE genes identified by the Halpern et al. study[12] for nine layers of mouse liver cells, we selected 94 genes detected in both human and mouse for correlation analysis. Expression values of each gene among the six clusters of human hepatocytes and nine layers of mouse liver cells were scaled and centered (separately in human and mouse) by z-scores. Finally, Pearson correlation was calculated using z-scores across all 94 genes to compare the six human hepatocytes clusters with the nine layers of mouse liver cells.

**Flow cytometry/intracellular cytokine staining**. Cell suspensions from TLH were stained as before[13,14] with live/dead aqua to assess viable cells and fluorophore-conjugated monoclonal antibodies to the following human cell-surface markers: anti-CD45-BV650 (Biolegend Clone: HI30, 1:20), anti-CD68-PE (Biolegend Clone: Y1/82 A, 1:20), anti-HLADR-AF700 (Biolegend Clone: L243, 1:20), and anti-MARCO (rabbit anti-human polyclonal) (Thermofisher: PA5-26888, 1:10); secondary donkey anti-rabbit-FITC (Invitrogen). Singlets were defined as having similar area and height measurements in forward scatter (FSC-A vs FSC-H). Gating strategy for cell-surface markers was set based on background auto-fluorescence measured in unstained controls (Supplementary Fig. 9). Intracellular cytokine staining. To examine functional differences in CD68$^+$ cells that were either MARCO$^+$ and MARCO$^-$, cell suspensions ($2 \times 10^6$ cells) from the NPC fraction were stimulated in 24-well plates with either 1 mg/ml LPS plus 25 ng/ml IFN-γ or 100 ng/ml LPS plus 25 ng/ml IL-1β for 18 h in the presence of BFA/monensin and intracellular secretion of TNF-α (detected with anti-human TNF-α antibodies; clone: MAb11, 1:20) was examined as previously described[13].

**Immunohistochemistry and immunofluorescence**. Human liver tissue was cut into 4 mm × 4 mm × 4 mm blocks and fixed in 10% formalin. 5–7 μM slides were cut from PFA or formalin-fixed, paraffin-embedded liver tissue resected from neurologically brain-dead donors. MARCO (Invitrogen, PA5-26888, 1:300) and CD68 (DAKO, PG-M1, 1:600) staining was performed on sequentially cut slides by the Toronto Pathology Research Program (Toronto General Hospital) using standard methods. Staining was performed with LT TE9 treated slides and donkey anti-rabbit secondary antibody conjugated to horseradish peroxidase. CD32B (Abcam, AB110076, 1:100) and CK19 (DAKO, MO888, 1:10) staining was performed on HIER (10 mM Citrate pH 6.0, 95 °C, 15 min) treated slides visualized by donkey anti-host (goat-AF488, mouse-AF555) secondaries and DAPI. Slides were scanned using the ScanScope AT2 (Lecia) at ×40 magnification by the Advanced Optical Microscopy Facility (AOMF) in Toronto. Lobules were defined by drawing a continuous line between portal triads around a single central vein using Halo software (Indica Labs, version). Each lobule was then concentrically partitioned into 10 layers between outer portal vein (layer 1) towards the central vein (layer 10). The positively stained area within each layer of the lobule (10 layers/lobule) was quantified and normalized to the area of the layer and presented as % positive stain. Ten or more individual lobules were defined. Representative partitioning of a lobule is shown in Supplementary Fig. 17. Lobule annotation were confirmed by a liver pathologist (O. Adeyi).

**Statistical analysis**. Statistical analysis for zonation figures was performed using GraphPad Prism v5.0. The utilized statistical test is listed in each figure caption. Statistical significance was evaluated using a one-way analysis of variance (ANOVA) with a Bonferroni post-test ***$P < 0.001$, **$P < 0.01$, *$P < 0.05$, ns = not significance ($P > 0.05$).

**Code availability**. GSEA Raw Data/ R Scripts for data process is available through https://github.com/BaderLab/singleLiverCells.

## Data availability

Sequence data that support the findings of this study (all Figures) is available through the NCBI GEO accession GSE115469. The analysed data is also available for viewing interactively as the R package HumanLiver, available at https://github.com/BaderLab/HumanLiver. The rest of the data is available from the authors upon reasonable request.

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

## Acknowledgements

This research was funded in part by the University of Toronto's Medicine by Design initiative, which receives funding from the Canada First Research Excellence Fund (CFREF) to IDM and by start-up funds from the Multi-Organ transplant program at the UHN to SAM. We would like to acknowledge Dr. M. Guilliams (VIB, Ghent University) and Dr. I.N. Crispe (University of Washington) for discussions of macrophage ontogeny. We would like to thank Joy Qu [www.joyqu.ca] for generating the illustration in Fig. 10. This work was supported by NRNB (U.S. National Institutes of Health, grant P41 GM103504) to GDB.

## Author contributions

S.A.M., J.C.L., G.D.B. and I.D.M designed the study. J.C.L., X.-Z.M., B.T.I., A.M.B., B.K. G., J.M., Z.S., N.K., N.W. and S.W.C., conducted experiments. E.L., J.E., I.L., N.S., M.S., A.G., D.G., P.G., G.S. and I.D.M harvested the surgical specimen. J.C.L., B.T.I. and G.D.B. wrote the analysis software. O.A. interpreted the histology. S.A.M., J.C.L., G.D.B., B.T.I., I.D.M., G.K., R.G., M.L.C., L.Y.L., D.C., R.K.S., S.O., M.O., M.D.L. and J.E.F. analyzed and interpreted the data. S.A.M., J.C.L., G.D.B. and I.D.M prepared the manuscript with critical revision from all authors.

## Additional information

**Competing Interests:** The authors declare no competing interests.

Sonya A. MacParland[1,2,3], Jeff C. Liu[4], Xue-Zhong Ma[1], Brendan T. Innes [4,5], Agata M. Bartczak[1], Blair K. Gage[6], Justin Manuel[1], Nicholas Khuu[7], Juan Echeverri[1], Ivan Linares[1], Rahul Gupta[1], Michael L. Cheng [3], Lewis Y. Liu[2], Damra Camat[1], Sai W. Chung[2], Rebecca K. Seliga[1], Zigong Shao[1], Elizabeth Lee[1], Shinichiro Ogawa[6], Mina Ogawa [6], Michael D. Wilson [5,8], Jason E. Fish[3,9], Markus Selzner[1], Anand Ghanekar[1], David Grant [1],

Paul Greig[1], Gonzalo Sapisochin[1], Nazia Selzner[1], Neil Winegarden [7], Oyedele Adeyi[1,3,10], Gordon Keller[6,11,12], Gary D. Bader [4,5] & Ian D. McGilvray[1]

[1]Multi-Organ Transplant Program, Toronto General Hospital Research Institute, Toronto, ON M5G 2C4, Canada. [2]Department of Immunology, University of Toronto, Toronto, ON M5S 1A8, Canada. [3]Department of Laboratory Medicine and Pathobiology, University of Toronto, Toronto M5G 1L7, Canada. [4]The Donnelly Centre, University of Toronto, Toronto, ON M5S 3E1, Canada. [5]Department of Molecular Genetics, University of Toronto, Toronto M5G 1A8, Canada. [6]McEwen Centre for Regenerative Medicine,  University Health Network, Toronto, ON M5G 1L7, Canada. [7]Princess Margaret Genomics Centre, University Health Network, Toronto, ON M5G 1L7, Canada. [8]Genetics and Genome Biology, Hospital for Sick Children, Toronto M5G 0A4, Canada. [9]Division of Advanced Diagnostics,  Toronto General Hospital Research Institute, Toronto, ON M5G 2C4, Canada. [10]Laboratory Medicine Program, University Health Network, Toronto, Ontario M5G 1L7, Canada. [11]Princess Margaret Cancer Centre, University Health Network, Toronto, Ontario M5G 1L7, Canada. [12]Department of Medical Biophysics, University of Toronto, Toronto, ON M5G 1L7, Canada. These authors contributed equally: Sonya A. MacParland, Jeff C. Liu, Xue-Zhong Ma. These authors jointly supervised the work: Gary D. Bader, Ian D. McGilvray.

