## [Peer Review File · Nature Communications]

Reviewers' comments:

Reviewer #1 (expert in Kupffer cells/liver immune cells)(Remarks to the Author):

This is an interesting and useful study but the paper suffers from over interpretation of the data and would be strengthened by complementary phenotyping and more localisation studies within intact tissue.

Major comments

The authors need to acknowledge that organ donors are not "normal" and provide more details on the age and background and medication of the donors as well as routine liver histology. Many donor livers will be from subjects with NAFLD for instance.

There are always concerns about collagenase digestion, however gentle. The authors should consider a comparison with non-enzymatic techniques, gentleMACs etc.

I am concerned that they are over interpreting the data. To talk about zonation they need to carry out more detailed immunolocalization of the cell types of interest. The assumptions on the distribution of HPCs seem very speculative without much more careful localization within intact tissue.

Cluster 14 hepatocytes were enriched in immune pathways but again it is difficult to extrapolate from this without evidence that these pathways are activated in zone 3 in intact tissue.

They describe the liver endothelial cells as LSEC but sinusoidal ECs are distinct and characterized by the presence of fenestrations and specific scavenger functions, they need phenotypic definition alongside the transcriptomics. The liver also contains venous and arterial and capillary vascular endothelium as well as lymphatic endothelium which may have strong similarities to SEC (Lalor 2013 Am J Physiol).

They need to be careful when talking about Kupffer cells which some authors would say only applies to yolk sac derived macrophages present in the fetal liver (Scultz Science 2012). I suggest they use the terms monocyte/macrophages. They did not report cells with a dendritic cell programme which is surprising and the heterogeneity of monocyte populations reported was not reflected in their findings. They must beware implying function from these data (ie tolerogenic KC etc).

The immune cell section requires much more careful phenotypic analysis to make sense of the data. Where are the ILCs, MAIT cells, NKTs and other atypical lymphocyte populations that we know are present in the human liver.

Reviewer #2 (expert in single cell RNA seq in liver)(Remarks to the Author):

In this manuscript, MacParland et al. profiled the single-cell transcriptomes of the human liver from 5 donors. 5 major cell types, including hepatocytes, endothelial cells, cholangiocytes, hepatic stellate cells and intrahepatic immune cells were identified. Subclusters were characterized and their marker gene expressions were discussed. This is a comprehensive profiling study of single-cell transcriptomes from adult human liver. However, some of the data presentations were lack of clarity. I have the following specific comments:

1. To better assess the quality of single-cell libraries, more sequencing details should be given. In particular, how is library size measured (Figures 2b, 4b, 5b)? What is the number of UMI detected per cell for cells in each cluster? What is the percentage of reads aligned to the transcriptomes, introns, ribosomes, and mitochondria? Is the sequencing saturated? How were doublets distinguished?
2. It is hard to correlate the cluster labels in Figure 2f with the labels used in Figures 3a, 3b. A consistent label system for different clusters should be used.
3. To understand the conservation of hepatocytes from mouse to human, a systematic comparison of gene expression of human hepatocytes in each zonation with corresponding cells from the mouse, as profiled in Halpern et al. (Single-cell spatial reconstruction reveals global division of labour in the mammalian liver, nature 2017), should be performed.
4. Shown in Figures 3a and 3b, different liver samples have vastly different representation in each

cell type. An alternative way to represent the data and illustrate donor to donor variation is to show a similar tSNE map as in Figure 1f, but coloring it by different donors. Some of the clusters, such as midzonal hepatocytes, are exclusively detected from liver 3. In the main text, the authors attributed this effect to be either from heterogeneity from donors or potential artifacts introduced during dissection. How is "midzone" anatomically defined? Importantly, this type of stratification is unlikely to be biological, especially midzonal hepatocytes should be relatively high abundant in all liver samples. How is batch effect controlled and normalized in the experiment? In addition, it is somewhat unexpected that hepatocytes do not cluster together. A pairwise comparison of hepatocytes from different clusters should also be performed to allow for a closer look at the differences between different hepatocyte clusters.

5. Judging from Figure 4c, hepatocyte marker such as ALB does not exclusively express in the clusters annotated as hepatocytes (very prominent in KC cluster, for example). Are those ALB expressions in cells from other clusters contamination? Similar issues are observed in other clusters. For example, PECAM1 is expressed in KC and Plasma cell clusters. Those cells with conflict marker expression should be examined carefully and taken out of downstream analysis if necessary.

6. The authors argue that hepatic stem cells do not form a discrete cluster in their data set. However, the only marker used in AFP (Figure 4c). Do the AFP+ cells also express other hepatic stem cell markers?

7. Pathway enrichment plots in Figures 4d, 4e, 4f, 5d, 8c are hard to interpret. The authors may consider showing the top enriched pathways by GSEA enrichment plots.

8. The finding of two distinct populations of Kupffer cells in the liver is intriguing. Orthogonal experiment is needed to confirm the differences of these two populations. The authors could consider sorting for MARCO+ vs MARCO- Kupffer cells and culture these cells and carry out functional assays to confirm that these two populations are functionally distinct; alternatively, the authors can perform immunofluorescent staining for additional markers that are differentially expressed between this two populations. In addition, how are these two populations compared with subpopulations previously identified in mice (See Kinoshita M, et al. Characterization of two F4/80-positive Kupffer cell subsets by their function and phenotype in mice. *J Hepatol.* 2010 and Zigmod E, et al. Infiltrating monocyte-derived macrophages and resident kupffer cells display different ontogeny and functions in acute liver injury. *J Immunol.* 2014)? Are CD11b, Ly6c also differentially expressed in these two populations?

9. Overall, the paper is very descriptive without detailed characterization nor validation of the markers for different cell types. Some aspects that the authors can further develop include: (1) Are the genes found in each cell type associated with known liver disease? (2) The authors may consider profiler livers with cirrhosis, hepatitis, liver cancers, for example, to see how the cell types and gene expression change during the disease process.

Reviewer #3 (expert Liver Heterogeneity) (Remarks to the Author):

This is an important paper that provides a comprehensive characterization of the cellular diversity in the human liver. The paper is the first study to perform single cell RNAseq of human livers, providing the gene expression signatures of the main cell types. Of particular note is the authors' optimized protocols to obtain a large number of viable human hepatocytes, a non-trivial task. The study is well performed and the paper is well written. The wealth of information provided here will be instrumental for many subsequent analyses of liver heterogeneity in normal and diseased livers.

Major comments

- Some of the cluster structure seems to depend on the patient and/or the numbers of UMIs. This can confound the interpretation of clusters as distinct cell types, especially for the hepatocytes, where some of the clusters only appear in one patient. The authors should use the Seurat functions to regress out both patients and numbers of UMIs, and then re-cluster the data. I believe this would provide a more robust and realistic clustering and may change some of the interpretations.

- Hepatocyte cluster 14 is denoted as Zone3 in Figure 2d, however the enriched pathways in Figure 4f rather show periportal functions (at least in mice) such as complement and clotting, the authors should discuss this discrepancy.

- The authors use an excessively high threshold for mitochondrial fraction, these cannot take up more than 30-40% of the cellular mRNA. The worry is that some damaged cells are included in the analysis, affecting the interpretation of the clusters. The authors should show that their main

conclusions are robust to the mitochondrial cutoff value and should add a supplementary figure coloring the tSNE dots by the mitochondrial RNA content.

- Cell cycle seems strange, hepatocytes are known to be significantly less proliferative than other cell types, e.g. cholangiocytes and endothelial cells, however Fig. S12 shows very similar G1 fractions, the authors should double check this. For example one could examine the expression of *Ccnd1* or other relatively highly expressed cyclin genes to see whether indeed hepatocytes have lower expression. The authors could also use the approach in PMID 27124452 (examine the summed expression of a few dozens of human cell cycle markers). In any case the authors should elaborate on which genes were used for cell cycle phasing.

- In Figure 10, B cells, T cells, plasma cells and NK cells are located to the portal part of the lobule, however no in-situ validation similar to the macrophage case of figure 8e are provided. This localization should either be backed up by validation or alternatively it should be stated that the location of these cell types is not representative of their zoned distribution.

Minor comments:

- The information gained from Fig. S14 is not clear.

- It could be nice to add some analysis of the rate of doublets, e.g. that fraction of cells are positive for e.g. both an endothelial cell marker and a hepatocyte marker.

- The authors should provide raw UMI tables.

- The authors should provide the full Seurat parameters for each analysis(i.e. clustering parameters), especially in the case of clusters 2, 8, 9 and 18 (line 461).

- Some of the information in Figure S1 and Figure 3 is redundant.

Point-By-Point Responses to Reviewer Comments:

Reviewer 1

- (1) This is an interesting and useful study but the paper suffers from over interpretation of the data and would be strengthened by complementary phenotyping and more localization studies within intact tissue.

We thank the reviewer for their helpful review. In response, we have carried out new experiments to strengthen our conclusions regarding liver macrophage phenotype, as presented in new **Figure 8d** and the new **Supplementary Figure 15**, and discussed in detail in the response to Comment 7 below. We have also strengthened our interpretation of how our hepatocyte clusters relate to zonation via pathway analysis (**new Figure 4D**) and a correlation analysis (**new Supplementary Figure 8**) with published mouse spatial gene expression patterns (Halpern et al., 2017). We have removed some interpretation of the data, particularly for cell types for which we have not yet performed tissue-specific localization studies (**see edited Figure 10**). We are currently optimizing methods to examine localization in the tissue (i.e. laser capture microdissection and single molecule FISH) but these are technically challenging and would unduly delay us sharing our results, thus we respectfully suggest are outside the scope of the current study.

- (2) *The authors need to acknowledge that organ donors are not "normal" and provide more details on the age and background and medication of the donors as well as routine liver histology. Many donor livers will be from subjects with NAFLD for instance.*

We completely agree with the reviewer that deceased donor human liver tissue is not "normal," and apologize if we did not emphasize that point enough in the original manuscript. Although we raised this issue in the original manuscript, we left the point to the Discussion section when discussing confounding factors (original Line 555: *"Possible confounding factors that should be considered when interpreting the data in the current study is that while the caudate lobes obtained were clinically acceptable healthy liver grafts, these deceased donor liver grafts are not "resting" – they have all been subjected to the systemic inflammation that accompanies brain death and are thus themselves inflamed [3]"*). In the revised manuscript, we raise the point sooner (**lines 100 and 178**). We also agree that fatty liver disease is quite common and could confound the results. The donor baseline characteristics, including age and race, are highlighted in **Supplementary Table 1**. As seen in this figure, only one donor liver (#3) was obtained from a donor with BMI>35. To further aid the interpretation of our data, we have now added representative histology figures for 4 of the 5 donor livers (**new Supplementary Figure 7**), where available. 4 of the 5 livers had biopsies taken, with the exception of #2. In the case of #2, the patient BMI was 31.7 with a low risk of fatty liver disease and no concerns raised during the organ retrieval

(complete BMI data in **Supplementary Table 1**). The available histology from all of the donor livers was reviewed by a clinical pathologist who specializes in liver (Author Dr. Oyedele Adeyi). The histology confirms that to the best of our knowledge none of our liver subjects had NAFLD: of note, the liver from Donor#3, with a BMI>35 and thus most at risk for NAFLD, had only 5-10%

fat on pathological review. We have commented on the histology in the revised manuscript on **lines 180-181**.

(3) There are always concerns about collagenase digestion, however gentle. The authors should consider a comparison with non-enzymatic techniques, gentleMACs etc.

We agree with the reviewer that any tissue manipulation, including enzymatic digestion, may impact the transcriptional faithfulness of our results. That said, we feel strongly that our methods of liver tissue dissociation are the most gentle and yield the most representative sample of cells possible. During the optimization of our methods, we found that without collagenase digestion, human and mouse liver dissociation is much less efficient, with significantly more cell death and in particular with a notable loss of parenchymal cells. Our impression is consistent with that of others in the field. Generally, mechanical protocols for hepatic tissue dispersion result in poor cellular viability and reduced macrophage recovery [4], both of which can be improved by enzymatic dispersal [5]. Morsy *et al.* compared a predominantly mechanical approach (scraping with a scalpel to dissociate), to a combined mechanical/enzymatic approach (gently chopping in the presence of enzymes), to the injection of enzymes into the vasculature with minimal mechanical isolation (most similar to our approach). These authors found that the mean viability for the mechanical approach was 39%, while the enzymatic/mechanical approach yielded a viability of 86.4% and the injected enzymes approach resulted in a 92.6% cellular viability. With these results in mind, we only carried out the combined approach and limited the length of time of collagenase digestion. It is worth noting that our method for gently dissociating the human liver took many years to develop, and is discussed in more detail in the reply to Comment #1 of Reviewer #3. We have shared this protocol on Protocols.io ([dx.doi.org/10.17504/protocols.io.m9sc96e](https://doi.org/10.17504/protocols.io.m9sc96e)).

We did find a gentleMACs protocol for mouse liver that has been optimized by Miltenyi Biotech; however, in this protocol the mechanical dissociation is still accompanied by enzymatic dissociation (please look at Datasheet in the following link: (<https://www.miltenyibiotec.com/CA-en/products/macs-sample-preparation/tissue-dissociation-kits/liver-dissociation-kit-mouse.html>))

We do plan to fully examine the impact of the dissociation protocol on the transcriptomic faithfulness of single cell profiles by comparing scRNA-seq to single nucleus RNA-seq (sNuc-seq)[6] from frozen tissue samples, avoiding proteolytic treatment and minimizing gene expression changes resulting from dissociation procedures. This method is still being developed in our genomics core facility so such an analysis will be carried out in a follow-up paper. This is discussed on lines **676-681** in the revised manuscript.

(4) I am concerned that they are over interpreting the data. To talk about zonation they need to carry out more detailed immunolocalization of the cell types of interest. The assumptions on the distribution of HPCs seem very speculative without much more careful localization within intact tissue.

We agree with the reviewer and apologize if we over-interpreted the data in the original manuscript, particularly vis-à-vis the human hepatocyte clusters. In the revised manuscript, we tried to limit any speculation, and instead described the similarities between our human hepatocyte clusters and the zoned hepatocyte gene expression carefully described in the mouse liver (Halpern *et al*, 2017), with additional support for our assertions from pathway analysis. As mentioned in the response to Comment 1, new **Supplementary Figure 8** shows the correlation of the genes expressed in our hepatocyte clusters to those of Halpern *et al*. 2017, who examined the spatial distribution of hepatocytes in mice. Corresponding changes to the manuscript are found on **lines 205-220**. A table showing the raw data represented in **Supplementary Figure 8** is included as **Supplementary Table 4**. We also performed a new pathway analysis examining all active pathways in each cluster (rather than just looking at unique pathways upregulated in each cluster). In the new analysis, rather than comparing significantly upregulated pathways in each cluster to pathways upregulated in the total hepatic cell population, we carried out a hepatocyte-specific analysis, comparing the hepatocyte clusters to the other hepatocyte clusters in order to determine most highly active cellular pathways for each hepatocyte cluster. To address this, we used GSVA analysis (gene set variation analysis package from Bioconductor). Specifically, we took all positively significant enriched cellular pathways for each cluster (cut-off at P value < 0.02 - this p-value was selected to show the most representative functions rather than all significant pathways). We have also attached the unfiltered GSVA raw data to support a deeper analysis by others (**Supplementary Table 5: GSVA raw data from Figure 4d**). In the pathway analysis, hepatocyte clusters were arranged according to their correlation with the zoned gene expression patterns described in mouse (**Supplementary Figure 8**). In new Figure 4d, we found that Cluster 5 (which correlates with periportal areas in mice) was enriched for liver cellular pathways characteristic of periportal function, including cholesterol metabolism and complement activation, along with numerous immune activation pathways (Fig. 4d). Meanwhile, pathway analysis revealed that the cells making up Clusters 1&3 (which correlate to central venous zones in mouse liver) were active in cellular pathways characteristic of zone 3 functions in mice and human including P450 pathways, drug metabolism, Wnt activation, hypoxia, amino acid (glutamine) biosynthesis, and glycolysis [7, 8], supporting the notion that these cells might have a central venous origin. The revised manuscript contains a new discussion of human hepatocyte zonation that takes into account the results of our correlation study, our new pathway analysis, and previously published human data using RNA profiling of the liver by laser capture microdissection and RNA-seq [8](**Lines 205-220**). For clarity, we made the statement that a complete characterization of the origin and identity of human hepatocytes will require additional examinations (**Lines 668-671**); with that caveat, our comparison of human to murine hepatocyte transcriptional profiles does support a correlation between individual hepatocyte clusters and sinusoidal zonation.

(5) Cluster 14 hepatocytes were enriched in immune pathways but again it is difficult to extrapolate from this without evidence that these pathways are activated in zone 3 in intact tissue.

The reviewer makes a good point. As now discussed in the revised manuscript we cannot make firm conclusions regarding zonation, since inferring zonation patterns of these clusters requires single-molecule FISH (smFISH) and laser capture microdissection of intact tissue. We have made this point clear in the revised manuscript on lines **205-208**. Any discussion of zonation has been made with respect to the Halpern et al., 2017 Nature paper in which spatial analyses in the mouse liver were carried out. We do observe clusters in our human liver scRNA-seq examinations that correspond to the mouse zones as defined by smFISH (**Supplementary Figure 8**), however, Cluster 14 showed no significant correlation with the mouse data. For Cluster 14, the top differentially expressed genes included many CYP genes, which originally led us to posit that these might be Zone 3 hepatocytes. However, the new pathway analysis presented in **revised Figure 4d** revealed that P450 pathways were not active in this cluster when compared to other hepatocyte clusters, and that functionally this cluster had more active cellular pathways that are found in periportal areas, such as complement activation in mouse [7], immune activation in human [8]. As well, this cluster was active in phosphatidyl metabolism and cholesterol efflux, which are linked to cholesterol and lipid homeostasis in mice [9]. As such, we have tempered our conclusions with respect to the possible zoned origin of this cluster. The new description of cluster 14 is found in the revised manuscript on **lines 265-274**.

(6) They describe the liver endothelial cells as LSEC but sinusoidal ECs are distinct and characterized by the presence of fenestrations and specific scavenger functions, they need phenotypic definition alongside the transcriptomics. The liver also contains venous and arterial and capillary vascular endothelium as well as lymphatic endothelium which may have strong similarities to SEC (Lalor 2013 Am J Physiol).

The reviewer makes a very important point: scRNA-seq data should be complemented, where possible, by phenotypic data. We have incorporated the Reviewer's points into our endothelial cell description, making reference to [10] on **lines 357-363**. However, we are currently limited in our ability to phenotype these cells. We have attempted to use human LSEC markers to facilitate live cell recovery (flow cytometry sorting CD45-CD68-CD32+). These studies, while still ongoing, have yielded purified populations but to date have not yet yielded high viability cells. We attempted to plate these cells but poor viability meant that the cells did not adhere. We cannot determine if this lack of viability is due to dissociation effects or subsequent deficits in culture conditions. Since rat LSECs only maintain fenestrations in culture for hours we believe that culture conditions may be the culprit although studies are ongoing in this area of our work. Also, it is worth noting that our analysis identifies several "endothelial" populations, two of which we have labelled as presumptive LSECs, and others as presumptive vascular endothelium. These points have been highlighted in **lines 357-363** in the revised manuscript.

(7) They need to be careful when talking about Kupffer cells which some authors would say only applies to yolk sac derived macrophages present in the fetal liver (Sultz Science 2012). I suggest they use the terms monocyte/macrophages. They did not report cells with a dendritic cell programme which is surprising and the heterogeneity of monocyte populations reported was not reflected in their findings. They must beware implying function from these data (ie tolerogenic KC etc).

We thank the reviewer for this comment, which raises important issues about cellular ontogeny and phenotyping. We have discussed these issues with our collaborator Dr. Martin Guilliams (macrophage expert at VIB, Ghent University). In the revised manuscript, we have related the expression profiles that characterize the two human liver macrophage populations to the mouse macrophage ontogeny literature. Our MARCO⁺ human liver macrophage population is most similar to long-lived, sessile, liver-resident murine Kupffer cells (either embryonic or monocyte derived) whereas our human liver inflammatory MACRO⁻ macrophages have a similar transcriptional profile to inflammatory, recently recruited macrophages as identified by two independent groups[2]. To further examine the issue of macrophage phenotype, we stimulated both populations of macrophages *in vitro* with endotoxin and IFN- γ , and examined cytokine secretion by intracellular cytokine staining. We found that MARCO positive macrophages secreted less TNF- α in response to LPS/ IFN- γ stimulation than MARCO negative liver resident CD68⁺ macrophages, suggesting that CD68⁺MARCO⁻ cells are more pro-inflammatory. MARCO positive cells also show enriched expression of *IL10* (Supplementary Table 2), as further evidence that these cells are immunoregulatory. The new data is presented in **Figure 8d**, and we have added description of these experiments to the methods (**Lines 812-817**) and have made mention in the results (**Lines 462-481**).

(8) The immune cell section requires much more careful phenotypic analysis to make sense of the data. Where are the ILCs, MAIT cells, NKTs and other atypical lymphocyte populations that we know are present in the human liver.

The reviewer raises a very important question that highlights some of the limitations of single cell RNA sequencing analysis for identifying low-frequency cell populations. We suspect that these cell populations are “hiding” in the larger lymphocyte clusters, and will become more clear as we and others add more liver donors and more cells to the analysis (from both healthy and diseased liver tissue). As we develop strategies to fractionate and enrich specific sub-populations of the cells we can also perform a much deeper analysis on these more rare cell types.

Our analysis does demonstrate that human liver-resident T cells have a unique identity compared to peripheral T cells: our study broadly outlines human “liver-resident” lymphocytes. As shown in original **Supplementary Figure 19**, the initial four clusters of T and NK-like cells can be sub-clustered into nine additional clusters that likely include ILCs, MAIT cells and NKTs. In the future, we plan to sequence greater numbers of cells from the total liver homogenate to more fully examine these populations. In the revised manuscript we emphasize that these more infrequent cell populations (for example, NKT cells are rare in healthy human liver, in contrast to mice[11]) are likely clustering within the four T/NK cell clusters and that we expect to uncover the unique identity of these populations with scRNAseq of greater numbers of cells. MAIT cells in particular are a subset of T cells with an $\alpha\beta$ TCR characterized by a semi-invariant TCR alpha (TCR α) chain. It is likely that these cells fall within Cluster 2, for example CD161 (*KLRB1*) is expressed on 52% of the cells in Cluster 2 (**Supplementary Table 2**). In the future, we will also employ TCR clonotyping to examine the transcriptional signature of CD161+ $\alpha\beta$ TCR+ cells expressing the known semi-invariant TCRs that are found in MAIT cells (*TRAV1-*

2/TRAJ12/20/3). Furthermore, we are developing panels of appropriate CITE-seq markers to improve our ability to detect and identify these important cell populations in our future analyses. In the meantime, we have altered the manuscript to reflect the fact that the T and NK-like clusters are likely multiple cell populations (**lines 560-561**). With this in mind, we have reclassified Cluster 2 as CD3+ $\alpha\beta$ TCR+ T cells and Cluster 8 as NK-like cells in all figures.

Reviewer #2 (expert in single cell RNA seq in liver)(Remarks to the Author):

(1) In this manuscript, MacParland et al. profiled the single-cell transcriptomes of the human liver from 5 donors. 5 major cell types, including hepatocytes, endothelial cells, cholangiocytes, hepatic stellate cells and intrahepatic immune cells were identified. Subclusters were characterized and their marker gene expressions were discussed. This is a comprehensive profiling study of single-cell transcriptomes from adult human liver. However, some of the data presentations were lack of clarity. I have the following specific comments:

We thank the reviewer for their supportive comment regarding the comprehensiveness of our study.

(2) To better assess the quality of single-cell libraries, more sequencing details should be given. In particular, how is library size measured (Figures 2b, 4b, 5b)?(number of reads in expression matrix) What is the number of UMI detected per cell for cells in each cluster? What is the percentage of reads aligned to the transcriptomes, introns, ribosomes, and mitochondria? Is the sequencing saturated? How were doublets distinguished?

We have addressed the question of sample quality control in new **Supplementary Figures 2,3 & 6**. To assess the sequence quality, we included the 10x Genomics Cell Ranger software summaries as an additional file (**Supplementary Information: Web summaries**). This data includes the percentage of reads aligned to the transcriptome, introns, and ribosomal sequences. The impact of various mitochondrial transcript ratio cut-offs is summarized in **Supplementary Figures 4 and 5**. The sequencing saturation ranged between 83.1% and 92.4%. The web summaries that we have included show the raw data across all cells profiled, live or dead. Since these summaries include both live and dead cells, there is a wide range of genes sequenced per cell in this data. As we mention in the approach, our purpose for sequencing all hepatic cells in the single cell suspensions is to minimize loss of cell types due to sample manipulation (**Supplementary Figure 1**). As seen in new **Supplementary Figure 2** ("Sample quality control") we show that after applying our filtering criteria to this raw data, the numbers of genes sequenced per cell is on average 1312 and falls within a reasonably tight range across all experiments (Range of 1148-1537).

In Figures 2b, 4b, 5b, we measure library size by determining the number of Unique Molecular Identifiers (UMIs) detected per cell, a measure or value which corresponds to the number of cDNAs reverse-transcribed in a droplet during the 10X Chromium method for sequencing library preparation.

Of note, we did not apply doublet filtering in part because we did not observe a natural threshold in library size per cell that we could choose to predict doublets. In addition, the doublet rate is expected to 0.9% per thousand cells. Given the number of cells we used per experiment (6000 cells were targeted) the doublet rate would be a maximum of 5%. Finally, there are naturally occurring binucleated hepatocytes in the liver [12] and we were concerned that it would be difficult to distinguish doublets and binucleated cells. Due to the heterogeneity of the liver tissue that we observe and the low number of doublets expected, it is unlikely that true doublets (with many possible cell type combinations) will result in a separate, unique cluster on a tSNE plot. In the new **Supplementary Figure 6** we show that most of the cells with the largest library sizes – the cells that might be predicted to be doublets - are concentrated in the Hepatocyte (Cluster #14) and plasma cells (Cluster #7). Plasmablasts, which traffic to the tissue[13] and binucleated hepatocytes[12] and have been previously described, suggesting that these are likely biological cell types and not doublets. We have made mention of this in the revised manuscripts **lines 160-174**.

(3) It is hard to correlate the cluster labels in Figure 2f with the labels used in Figures 3a, 3b. A consistent label system for different clusters should be used.

We have modified **Figure 3** to reflect the clusters in **Figure 2f** using a modified naming, based on the Reviewer's suggestions. In our modified naming of the clusters, we have reframed our discussion of possible zonation of hepatocytes in response to **Reviewer 1 Comments 1, 4 & 5**, since we cannot make firm conclusions about hepatocyte zonation based on our data. As such, we have changed the hepatocyte labels in **Figure 2f** and **3**.

(4) To understand the conservation of hepatocytes from mouse to human, a systematic comparison of gene expression of human hepatocytes in each zonation with corresponding cells from the mouse, as profiled in Halpern et al. (Single-cell spatial reconstruction reveals global division of labour in the mammalian liver, nature 2017), should be performed.

Thank you for this valuable suggestion. In our **new Supplementary Figure 8**, we compare the gene expression patterns of our human hepatocyte clusters to those in the mouse liver zones profiled by Halpern *et al.* [7]. Of all one-to-one orthologous genes defined by the Ensembl database comparing human and mouse, 94 were significantly zoned marker genes defined by the mouse paper. We correlated our six human hepatocyte clusters with the nine layers of mouse liver cells defined by Halpern *et al.* using the expression levels of these 94 genes. As shown in the heat map of new **Supplementary Figure 8**, this analysis revealed that the gene expression patterns in four of the human hepatocyte clusters (1,3,5, and 15) correlated significantly with at least one layer of the zoned gene expression patterns identified in the mouse sinusoid (Supplementary Figure 8). Two clusters (6 and 14) showed weaker correlation, which may be due to differences in the genes that define mouse and human zoned liver expression patterns. Specifically, human Cluster 5 correlated best with the most periportal mouse liver layer, while human Cluster 3 correlated best with the most central venous mouse

liver layer. Human Cluster 1 was correlated with mouse layers 2 and 3 (more periportal) and Cluster 15 correlated best with an interzonal mouse layer (Layer 4). Clusters 6 and 14 did not correlate significantly with any mouse layer, though the overall correlation pattern of Cluster 14 suggests zonation similar to Cluster 5. The weaker correlation for these clusters may be due to lack of overlap of differentially expressed genes defining these clusters and the top 94 genes which defined zonation in mouse liver. We have described this in the manuscript (**results line 205-227**). Although the correlations suggest that the clusters may be zoned, we cannot definitively assign a clear zone to these cells without additional spatial examinations in human liver tissue. We have modified the hepatocyte section to reflect this caveat (**results line 669-681**).

(5) Shown in Figures 3a and 3b, different liver samples have vastly different representation in each cell type. An alternative way to represent the data and illustrate donor to donor variation is to show a similar tSNE map as in Figure 1f, but coloring it by different donors. Some of the clusters, such as midzonal hepatocytes, are exclusively detected from liver 3. In the main text, the authors attributed this effect to be either from heterogeneity from donors or potential artifacts introduced during dissection. How is midzone; anatomically defined? Importantly, this type of stratification is unlikely to be biological, especially midzonal hepatocytes should be relatively high abundant in all liver samples. How is batch effect controlled and normalized in the experiment? In addition, it is somewhat unexpected that hepatocytes do not cluster together. A pairwise comparison of hepatocytes from different clusters should also be performed to allow for a closer look at the differences between different hepatocyte clusters.

As suggested, in **new Figure 3b**, we have added a tSNE map with the donors depicted using different colors. In this Figure, we again see that clusters of hepatocytes, particularly clusters 1,3, 5 and 6, were attributed to patient 2 or 3. We have preserved the original stacked bar plot of donor variability (**Figure 3a**) because cells on the tSNE plot may be plotted on top of each other, obfuscating the coloring by donor (e.g. the impression given from the tSNE plot is that Cluster 16 derives almost entirely from donor 5 (purple), but it is actually only 60% from this donor). We did try to correct batch effects between donors in early analysis, but were not able to identify any obvious effect. In particular, we observed excellent overlap of most cell types across donors (suggesting no overall batch effect), and only strong donor specific effects in a few cell types. We have now extensively re-examined the patient/batch-specific structure in the data. We regressed out technical factors including donor, library size, and gene detection rate. The non-hepatocyte clusters were very robust to this correction, indicating that the biological signal separating cell types in the clustering analysis was sufficient to overcome technical variation within each cell type. The donor-specific differences between hepatocyte clusters persisted after this correction. Our interpretation of these findings is that hepatocytes are particularly sensitive to the effects of dissociation. This topic is discussed in depth in the response to **Reviewer 3**, major **Comment 2**.

(6) Judging from Figure 4c, hepatocyte marker such as ALB does not exclusively express in the clusters annotated as hepatocytes (very prominent in KC cluster, for example). Are those ALB expressions in cells from other clusters contamination? Similar issues are observed in other clusters. For example, PECAM1 is expressed in KC and Plasma cell clusters. Those cells with conflict marker expression should be examined carefully and taken out of downstream analysis if necessary.

The reviewer brings up an important question that relates to the cell-specificity of gene and surface marker expression. We suspect that our results are more reflective of heterogeneous gene/marker expression than cluster contamination. As described in the response to **Reviewer 3 (Comment 8)**, possible doublets (**seen in new Supplementary Figure 6**) were mainly detected in Cluster 14, and Cluster 7 suggesting that the CD31 and ALB detected was likely not due to contamination from other clusters. CD31 may not be endothelial cell specific [14], and has been described as being expressed on plasma cells [15] and macrophages [16]. ALB expression has also been described in cell types other than hepatocytes including mouse liver resident macrophages [17] and human myelomonocytic cells [18], and may be a marker of a subset of human liver macrophages. In the future, we plan to carry out sNuc-Seq on stored frozen samples from these same livers to further examine the specificity of these signals. This is discussed on **lines 676-681** in the revised manuscript.

(7) The authors argue that hepatic stem cells do not form a discrete cluster in their data set. However, the only marker used in AFP (Figure 4c). Do the AFP+ cells also express other hepatic stem cell markers?

A discrete stem cell cluster would be a very useful to identify, though it may also have been obscured as a rare cell population (also see response to **Reviewer 1, Comment 8**). In attempting to further define the AFP⁺ cells, we have now carried out a pair-wise analysis of the top enriched cellular pathways in all AFP expressing cells. As suggested, we selected any hepatocytes with detectable AFP expression and generated a ranked list of genes that were differentially expressed in AFP⁻ vs AFP⁺ hepatocytes (**Supplementary Table 6**). We then used the ranked gene list to examine enriched cellular pathways cells in the AFP expressing cells using pathway analysis. The results are shown in **new Supplementary Figure 10**. We found enriched pathways in AFP⁺ cells including those for cellular division and IL-6/7 signaling. This finding supports the assertion that these cells may be hepatic stem cells since IL-6 is a key cytokine for the proliferation of hepatocytes [19]. These findings are discussed in the revised manuscript (**lines 294-301**).

(8) Pathway enrichment plots in Figures 4d, 4e, 4f, 5d, 8c are hard to interpret. The authors may consider showing the top enriched pathways by GSEA enrichment plots.

We apologize for the lack of clarity in pathway analysis in the original manuscript. We have expanded our explanation of the pathway analysis in the legend for **Figure 5d** and have re-presented the pathway analysis for **Figure 8c** to try to improve clarity. We are using the standard Enrichment Map visualization that helps summarize the entire pathway analysis and is implemented in GSEA. The GSEA enrichment plots are only useful for one pathway at a time,

but we are presenting hundreds of pathways. Our enrichment plots shown in Figures. **5d** and **8c** integrate the data from the GSEA enrichment plots [20] to make the GSEA information easier to interpret. The strength of the enrichment is mapped to color to make the gradient more informative. In our enrichment analysis, we only present the results that we are confident of (specifically, we present all significant uniquely enriched cellular pathways for each cluster (cut-off at P value < 0.01); however, we have provided a link to the full GSEA enrichment analysis (<https://github.com/drseamster/singleLiverCellRNA>) for each of the pathway analysis Figures (**Figs. 5d, 8c and Supplementary Figures 10, 12 and 13**) to support a deeper analysis by others.

We have also replaced the original hepatocyte pathway analysis in original **Figures 4d, 4e** and **4f** with a new pathway analysis showing the active pathways in each of the clusters (**new Figure 4d**). In the new analysis, rather than comparing significantly upregulated pathways in each cluster to pathways upregulated in the total hepatic cell population, we carried out a hepatocyte-specific analysis, comparing the hepatocyte clusters to the other hepatocyte clusters in order to determine most highly active cellular pathways for each hepatocyte cluster (described in response to **Reviewer 1, Comment 4**). These analyses more clearly show the pathways upregulated in the hepatocyte clusters (the groups determined based on the correlational analysis suggested by the reviewers and shown in **Supplementary Figure 8**) and discussed on **lines 221- 227, 237-240, 253-257,278-281, 781-790** in the revised manuscript.

(9) The finding of two distinct populations of Kupffer cells in the liver is intriguing. Orthogonal experiment is needed to confirm the differences of these two populations. The authors could consider sorting for MARCO+ vs MARCO- Kupffer cells and culture these cells and carry out functional assays to confirm that these two populations are functionally distinct; alternatively, the authors can perform immunofluorescent staining for additional markers that are differentially expressed between this two populations. In addition, how are these two populations compared with subpopulations previously identified in mice (See Kinoshita M, et al. Characterization of two F4/80-positive Kupffer cell subsets by their function and phenotype in mice. J Hepatol. 2010 and Zigmod E, et al. Infiltrating monocyte-derived macrophages and resident kupffer cells display different ontogeny and functions in acute liver injury. J Immunol. 2014)? Are CD11b, Ly6c also differentially expressed in these two populations?

The reviewer points out the importance of bringing together supportive functional and correlative data in interpreting scRNA-seq data; our response is outlined above, in reply to **Comment 7** by **Reviewer 1**. In terms of CD11b and Ly6c, we examined the expression of CD11b (*ITGAM*) and Ly6C (*CD59*) and found that neither were defining genes in the human liver macrophage clusters although we acknowledge that it is possible these particular genes are not detected at the RNA level in scRNA-seq but may have been present at the protein level. Below are tSNE plots showing the expression of *ITGAM* and *CD59* in our 20 clusters. In the Kinoshita study [4], CD11b characterized the inflammatory macrophage population. The lack of CD11b (*ITGAM* expression) as a defining characteristic of inflammatory macrophages in our study can be attributed to differences between human and mice. In the study by Zigmond et al. [21] genes such as *MARCO* and *CD163* were highly expressed on Kupffer cells in steady state, and down-regulated in infiltrating macrophages during acute liver injury. We have made mention of

the comparison between the human macrophage populations we identified compared to previously described mouse intrahepatic macrophages in the manuscript (**results line 462-481**).

(10) Overall, the paper is very descriptive without detailed characterization nor validation of the markers for different cell types. Some aspects that the authors can further develop include: (1) Are the genes found in each cell type associated with known liver disease? (2) The authors may consider profiler livers with cirrhosis, hepatitis, liver cancers, for example, to see how the cell types and gene expression change during the disease process.

We completely agree with the reviewer that the studies that follow need to examine the roles of the genes and cell types we have described in the context of liver disease.

Some supportive work has already been done by others, though not usually in the context of liver disease. For example, *MARCO*, a gene which differentiates our liver macrophage populations, has been examined in studies of cancer progression and cancer targeting. The expression of *MARCO* in the tumor microenvironment has been linked to worse outcome in human breast cancer [22]. *MARCO* has also been examined in preclinical mouse models in the context of using *MARCO* to target cancer development and metastasis [23]. In this work, *MARCO* expression defined a subtype of suppressive tumor associated macrophages (TAMs). The authors also employed anti-*MARCO* antibody and reprogrammed TAMs towards a more inflammatory phenotype that had increased tumor immunogenicity. They also found that anti-*MARCO* mAb enhanced the efficacy of checkpoint therapy anti-CTLA4 in a mouse colon cancer and melanoma model. A description and this reference have been added to the manuscript on **lines 474-481**.

However, profiling diseased liver tissue is the next step in clearly identifying the cellular drivers of liver disease, and we are in the process of gearing up for these studies. Profiling diseased liver tissue in a manner that limits dissociation bias is one of our current focusses. We are now

examining the impact of enzymatic and non-enzymatic dissociation on the sorts of biopsies and explants that we have access to for a diseased liver study – we feel that this optimization work is beyond the scope of the present study. The present study is intended as a baseline for a large number of impactful studies into the basis of liver disease. In that regard, we are in the process of establishing a collaborative, multi-national research group to perform scRNA-seq studies in the context of liver disease, and will share our data, protocols and approaches with this community. We would respectfully submit that the scale of this question (“cellular drivers of liver disease”) – and of the work involved – is vast, and will take many years to complete. That said, as pointed out by **Reviewer 3**, the work presented in the current study will serve as the necessary first step.

Reviewer #3 (expert Liver Heterogeneity) (Remarks to the Author):

(1) This is an important paper that provides a comprehensive characterization of the cellular diversity in the human liver. The paper is the first study to perform single cell RNAseq of human livers, providing the gene expression signatures of the main cell types. Of particular note is the authors' optimized protocols to obtain a large number of viable human hepatocytes, a non-trivial task. The study is well performed and the paper is well written. The wealth of information provided here will be instrumental for many subsequent analyses of liver heterogeneity in normal and diseased livers.

We thank the reviewer for the supportive comments that highlight the challenges of isolating viable hepatocytes from human liver tissue. Indeed, we found this to be a difficult task which took many years to optimize. In our experience, the recovery of hepatocytes is directly related to the completeness of the liver perfusion (for dissociation) and to the amount of vasculature in the retrieved tissue that was available for cannulation. We started optimizing the handling of human liver tissue six years ago and have considered and modified the following: **1)** catheter size (1.2-2mm diameter olive-tipped irrigation cannulae: manufacturer: Ernst Kratz GmbH cat no: 1464LL, 1465LL); **2)** perfusion solution; **3)** duration of perfusion; **4)** source of digestion enzymes; **5)** speed of perfusion (10 ml/min/cannulae); **6)** methods of securing the cannula (we employ sutures and vetbond glue); **7)** temperature and oxygenation at each step (this has greatly improved the post-dissociation viability); **8)** minimizing steps in the warm – 37°C – to minimize cellular activation. We have made this protocol a resource on protocols.io (dx.doi.org/10.17504/protocols.io.m9sc96e) (highlighted in manuscript on **line 100**), and have expanded on the protocol in the revised manuscript (**lines 695-701**).

Major comments

(2) Some of the cluster structure seems to depend on the patient and/or the numbers of UMIs. This can confound the interpretation of clusters as distinct cell types, especially for the hepatocytes, where some of the clusters only appear in one patient. The authors should use the Seurat functions to regress out both patients and numbers of UMIs, and then re-cluster the data. I believe this would provide a more robust and realistic clustering and may change some of the interpretations.

We thank the reviewer for this helpful suggestion. The revised manuscript includes a tSNE plot showing the contribution of each individual donor to the clusters (**New Figure 3B**). As seen in the figure, while some of the hepatocyte clusters appear only in one liver, the immune cell and endothelial cell clusters are found in all five livers. We attribute these differences both to donor-to-donor differences and to dissociation effects. The fact that the immune cell and monocyte/ macrophage populations from all donors overlap nicely argues that our conclusions about the presence of distinct monocyte/macrophage populations in the liver is robust.

To address the patient/batch-specific structure in the hepatocyte clusters, we attempted multiple strategies for batch correction. As suggested by the reviewer, we regressed out technical factors including donor, library size, and gene detection rate. The non-hepatocyte clusters were very robust to this correction, indicating that the biological signal separating cell types was sufficient to overcome technical variation within each cell type. The donor-specific differences between hepatocyte clusters persisted after this correction. We also applied more complex batch correction methods, specifically the canonical correlation analysis-based method implemented in Seurat (doi:10.1038/nbt.4096) and the mutual nearest neighbours method implemented in scran (doi:10.1101/165118) to the entire dataset, with similar results. Given that the batch effect was concentrated in the hepatocytes, and donors 2 and 3 were the primary contributors of hepatocytes, we further attempted the above corrections on only the hepatocyte clusters, and then on only the hepatocytes contributed by donors 2 and 3, with similar results. Consequently, while we are not comfortable speculating on the source of the liver-specific differences between hepatocyte clusters, we feel that they represent a true biological signal present in the data. It is very likely that there is donor-to-donor variability in hepatocytes, a finding that we intend to examine more closely in the future by examining more human livers. We have addressed these limitations in the revised manuscript (**lines 194-204 & 671-674**).

Our experience and data suggest that the human hepatocyte is extremely fragile and susceptible to mechanical and chemical injury during the process of liver cell isolation – and that these cells are injured rather than lost during this process. We generally obtain ($0.5-1 \times 10^8$ cells/g liver tissue), but the number of cells recovered varies depending on the completeness of the dissociation. With every liver dissociation we plate the hepatocytes, and we have found that hepatocyte yield and viability (as evidenced by plating and spreading of the hepatocytes) is directly related to the number of vessels available to cannulate the liver tissue and the completeness of perfusion (as indexed by the time it takes during the enzymatic digestion for the caudate lobe to lose its stiffness). By contrast, we obtain non-parenchymal/immune cells from all isolations. Thus, we expect that data regarding these latter cellular subsets can be rapidly accrued and compared across institutions once a clear protocol is agreed upon. However, as we move forward in defining the human hepatocytes, great care will need to be taken to describe the nature and results of the hepatocellular isolation for the results to be comparable.

(3) Hepatocyte cluster 14 is denoted as Zone3 in Figure 2d, however the enriched pathways in Figure 4f rather show periportal functions (at least in mice) such as complement and clotting, the authors should discuss this discrepancy.

We thank the reviewer for this comment, which prompted us to re-examine both our classification of this cluster as well as our pathway analysis. The top differentially expressed genes in this cluster included many CYP genes. However, the revised pathway analysis presented in **revised Figure 4d** revealed that P450 pathways were not uniquely active in this cluster, compared to other hepatocyte clusters, and that functionally this cluster had more pathways that are found in periportal areas (e.g. complement activation in mouse[7], immune activation in human [8]). As well, this cluster was active in phosphatidyl metabolism and cholesterol efflux, which are linked to cholesterol and lipid homeostasis in mice[9]. To further explore the identity of this cluster, we examined whether the most highly differentially expressed genes in this cluster correlated with zonated genes expressed in mouse liver and found no significant correlation. As such, we have tempered our conclusions with respect to the possible zonated origin of this cluster. The new description of cluster 14 is found in the revised manuscript on **lines 265-274**.

(4) The authors use an excessively high threshold for mitochondrial fraction, these cannot take up more than 30-40% of the cellular mRNA. The worry is that some damaged cells are included in the analysis, affecting the interpretation of the clusters. The authors should show that their main conclusions are robust to the mitochondrial cutoff value and should add a Supplementary figure coloring the tSNE dots by the mitochondrial RNA content.

We apologize if the original manuscript was not clear regarding our rationale for our mitochondrial fraction threshold. We had included a higher mitochondrial transcript threshold due to the high mitochondrial activity in the liver. In order to address whether this threshold was appropriate, we have performed new analyses as presented in **Supplementary Figures 4 and 5**. We tested the robustness of our analysis by reanalyzing the data after setting mitochondrial cutoffs from 10-60% of the cellular mRNA. As seen in new **Supplementary Figure 4**, all mitochondrial cutoffs (0.1, 0.2, 0.3, 0.4, 0.5, and 0.6), result in cells from all 20 clusters being identified. All clusters (except cluster #6 at 0.1 cutoff) are visually identified as unique populations in tSNE plots at all cutoffs. Furthermore, the additional cells from mitochondrial cut-off 0.6 to 0.3 are found in almost all clusters, thus are not cell type biased (new **Supplementary Figure 5**). Altogether, our results indicate that the cell clusters identified at the original 0.5 cutoff are robust and consistent, thus we maintain this threshold in our analysis. We have added a comment to this effect in the revised manuscript, results line **148-155**.

(4) Cell cycle seems strange, hepatocytes are known to be significantly less proliferative than other cell types, e.g. cholangiocytes and endothelial cells, however Fig. S12 shows very similar G1 fractions, the authors should double check this. For example, one could examine the expression of Ccnd1 or other relatively highly expressed cyclin genes to see whether indeed hepatocytes have lower expression. The authors could also use the approach in PMID 27124452 (examine the summed expression of a few dozens of human

cell cycle markers). In any case the authors should elaborate on which genes were used for cell cycle phasing.

We appreciate the reviewer's suggestion, and in the revised manuscript have compared our original predictions to those of an alternative cell-cycle phase prediction method implemented in Seurat. Seurat's method scores each cell based on its expression of a set of canonical G2/M and S-phase marker genes (see below for gene lists). We find that the cell cycle phase prediction from Seurat aligns more closely with expected biology in that hepatocytes are proliferating less than the immune cell subsets. The results are shown in new **Supplementary Figure 18**.

Gene list for cell cycle from Seurat:

S: *MCM5, PCNA, TYMS, FEN1, MCM2, MCM4, RRM1, UNG, GINS2, MCM6, CDCA7, DTL, PRIM1, UHRF1, MLF1IP, HELLS, RFC2, RPA2, NASP, RAD51AP1, GMNN, WDR76, SLBP, CCNE2, UBR7, POLD3, MSH2, ATAD2, RAD51, RRM2, CDC45, CDC6, EXO1, TIPIN, DSCC1, BLM, CASP8AP2, USP1, CLSPN, POLA1, CHAF1B, BRIP1, E2F8*

G2/M: *HMGB2, CDK1, NUSAP1, UBE2C, BIRC5, TPX2, TOP2A, NDC80, CKS2, NUF2, CKS1B, MKI67, TMPO, CENPF, TACC3, FAM64A, SMC4, CCNB2, CKAP2L, CKAP2, AURKB, BUB1, KIF11, ANP32E, TUBB4B, GTSE1, KIF20B, HJURP, CDCA3, HN1, CDC20, TTK, CDC25C, KIF2C, RANGAP1, NCAPD2, DLGAP5, CDCA2, CDCA8, ECT2, KIF23, HMMR, AURKA, PSRC1, ANLN, LBR, CKAP5, CENPE, CTCF, NEK2, G2E3, GAS2L3, CBX5, CENPA.*

(5) In Figure 10, B cells, T cells, plasma cells and NK cells are located to the portal part of the lobule, however no in-situ validation similar to the macrophage case of figure 8e are provided. This localization should either be backed up by validation or alternatively it should be stated that the location of these cell types is not representative of their zoned distribution.

We certainly agree that tissue-specific localization studies complement scRNA-seq data. While it has been shown that these cells display zoned distribution with a relative accumulation in the portal regions of the sinusoids [24], we have not confirmed this in the present study. We plan to do so in the future, along with studies of the roles of these cells in liver disease. We have now stated clearly in the revised figure legend that the location of these cells types reflects consensus pathological localization from prior work and is not directly demonstrated by our data (Fig. 10 legend, line 7).

Minor comments:

(6) The information gained from Fig. S14 is not clear.

We agree with the reviewer and have removed this heatmap.

(7) It could be nice to add some analysis of the rate of doublets, e.g. that fraction of cells are positive for e.g. both an endothelial cell marker and a hepatocyte marker.

This is an important question that we have addressed in **new Supplementary Figure 6** – also see response to **reviewer 2, comment 2**. We did not apply doublet filtering because: 1. we did not observe a natural threshold in library size per cell that we could choose to predict doublets; 2. the doublet rate is expected to be low given the number of cells we used per experiment based on 10x Genomics information; and 3. there are naturally occurring binucleated hepatocytes in liver [12] and we were concerned that it would be difficult to distinguish doublets and true binucleated cells. Due to the heterogeneity of the liver tissue that we observe and the low number of doublets expected, it is unlikely that true doublets (with many possible cell type combinations) will result in a separate, unique cluster on a tSNE plot. In **new Supplementary Figure 6** we show that most of the cells with the largest library sizes, that may be predicted to be doublets, concentrated in the Hepatocyte (Cluster #14) and plasma cell (Cluster #7) populations suggesting that these are likely biological cells and not doublets. We have commented on this new analysis in the revised manuscript on **lines 164-169**.

(8) The authors should provide raw UMI tables.

The RAW data has now been submitted to GEO. Accession numbers are presently being assigned.

(9) The authors should provide the full Seurat parameters for each analysis (i.e. clustering parameters), especially in the case of clusters 2, 8, 9 and 18 (line 461).

We have included the full Seurat parameters in the Figure Legends.

(10) Some of the information in Figure S1 and Figure 3 is redundant.

On reflection, we agree with the reviewer; we have removed Supplementary Figure 1C

REFERENCES:

1. **Gibbins SL, Goyal R, Desch AN, Leach SM, Prabagar M et al.** Transcriptome analysis highlights the conserved difference between embryonic and postnatal-derived alveolar macrophages. *Blood* 2015;126(11):1357-1366.
2. **Scott CL, Zheng F, De Baetselier P, Martens L, Saey Y et al.** Bone marrow-derived monocytes give rise to self-renewing and fully differentiated Kupffer cells. *Nat Commun*, Article 2016;7:10321.
3. **Borozan I, Chen L, Sun J, Tannis LL, Guindi M et al.** Gene expression profiling of acute liver stress during living donor liver transplantation. *American journal of transplantation : official journal of the American Society of Transplantation and the American Society of Transplant Surgeons* 2006;6(4):806-824.
4. **Kinoshita M, Uchida T, Sato A, Nakashima M, Nakashima H et al.** Characterization of two F4/80-positive Kupffer cell subsets by their function and phenotype in mice. *J Hepatol* 2010;53(5):903-910.

5. **Morsy MA, Norman Pj Fau - Mitry R, Mitry R Fau - Rela M, Rela M Fau - Heaton ND, Heaton Nd Fau - Vaughan RW et al.** Isolation, purification and flow cytometric analysis of human intrahepatic lymphocytes using an improved technique. (0023-6837 (Print)).
6. **Habib N, Avraham-Davidi I, Basu A, Burks T, Shekhar K et al.** Massively parallel single-nucleus RNA-seq with DroNc-seq. *Nat Methods* 2017;14(10):955-958.
7. **Halpern KB, Shenhav R, Matcovitch-Natan O, Toth B, Lemze D et al.** Single-cell spatial reconstruction reveals global division of labour in the mammalian liver. *Nature*, Article 2017;542(7641):352-356.
8. **McEnerney L, Duncan K, Bang BR, Elmasry S, Li M et al.** Dual modulation of human hepatic zonation via canonical and non-canonical Wnt pathways. *Exp Mol Med*, Original Article 2017;49(12):e413.
9. **Scapa EF, Pocai A, Wu MK, Gutierrez-Juarez R, Glenz L et al.** Regulation of energy substrate utilization and hepatic insulin sensitivity by phosphatidylcholine transfer protein/StarD2. *FASEB J* 2008;22(7):2579-2590.
10. **Lalor PF, Herbert J, Bicknell R, Adams DH.** Hepatic sinusoidal endothelium avidly binds platelets in an integrin-dependent manner, leading to platelet and endothelial activation and leukocyte recruitment. *Am J Physiol Gastrointest Liver Physiol* 2013;304(5):G469-478.
11. **Kenna T, Golden-Mason L, Porcelli SA, Koezuka Y, Hegarty JE et al.** NKT cells from normal and tumor-bearing human livers are phenotypically and functionally distinct from murine NKT cells. *J Immunol* 2003;171(4):1775-1779.
12. **Duncan AW, Taylor MH, Hickey RD, Hanlon Newell AE, Lenzi ML et al.** The ploidy conveyor of mature hepatocytes as a source of genetic variation. *Nature* 2010;467(7316):707-710.
13. **Seong Y, Lazarus NH, Sutherland L, Habtezion A, Abramson T et al.** Trafficking receptor signatures define blood plasmablasts responding to tissue-specific immune challenge. *JCI Insight* 2017;2(6):e90233.
14. **Liu L Fau - Shi G-P, Shi GP.** CD31: beyond a marker for endothelial cells. (1755-3245 (Electronic)).
15. **Govender D, Harilal P, Dada M, Chetty R.** CD31 (JC70) expression in plasma cells: an immunohistochemical analysis of reactive and neoplastic plasma cells. *J Clin Pathol* 1997;50(6):490-493.
16. **McKenney JK, Weiss SW, Folpe AL.** CD31 expression in intratumoral macrophages: a potential diagnostic pitfall. *Am J Surg Pathol* 2001;25(9):1167-1173.
17. **Lavin Y, Winter D, Blecher-Gonen R, David E, Keren-Shaul H et al.** Tissue-resident macrophage enhancer landscapes are shaped by the local microenvironment. *Cell* 2014;159(6):1312-1326.
18. **Thiel CS, Hauschild S, Tauber S, Paulsen K, Raig C et al.** Identification of reference genes in human myelomonocytic cells for gene expression studies in altered gravity. *Biomed Res Int* 2015;2015:363575.
19. **Schmidt-Arras D, Rose-John S.** IL-6 pathway in the liver: From physiopathology to therapy. (1600-0641 (Electronic)).
20. **Merico D, Isserlin R, Stueker O, Emili A, Bader GD.** Enrichment map: a network-based method for gene-set enrichment visualization and interpretation. *PLoS One* 2010;5(11):e13984.

21. **Zigmond E, Samia-Grinberg S, Pasmanik-Chor M, Brazowski E, Shibolet O et al.** Infiltrating monocyte-derived macrophages and resident kupffer cells display different ontogeny and functions in acute liver injury. *J Immunol* 2014;193(1):344-353.
22. **A B, E T, T S, B N, T T et al.** - Extracellular matrix signature identifies breast cancer subgroups with different. *D - 0204634 (- 0022-3417 (Print))*:- 357-367.
23. **Georgoudaki AM, Prokopec KE, Boura VF, Hellqvist E, Sohn S et al.** Reprogramming Tumor-Associated Macrophages by Antibody Targeting Inhibits Cancer Progression and Metastasis. *Cell Rep* 2016;15(9):2000-2011.
24. **Doherty DG, O'Farrelly C.** Innate and adaptive lymphoid cells in the human liver. *Immunological Reviews* 2000;174(1):5-20.

Reviewers' comments:

Reviewer #1 (Remarks to the Author):

Thank you for your thoughtful and comprehensive responses to my criticisms. I am happy that my concerns have wherever practical been addressed. This is an important and extremely useful paper.

Reviewer #2 (Remarks to the Author):

In the revised manuscript, the authors included more in-depth single-cell sequencing QC matrices, reformatted some of the figures, compared the human hepatocytes single-cell RNA-seq data from landmark genes from mouse, and clarified some of the data interpretations. Overall, this results in a much-improved version of the manuscript. I have the following remaining questions/comments:

1. Page 3, there are multiple single-cell studies on the human pancreas (Please note that all the references, including the one already cited in the manuscript, used isolated human islets as starting material). Please add in the following references:

- (1) Li J, Klughammer J, et al. EMBO Rep. 2016 Feb;17(2):178-87.
- (2) Wang YJ, Shug J, et al. Diabetes 2016 Oct; 65(10): 3028-3038.
- (3) Xin Y, Kim J, et al. Cell Metab. 2016 Oct 11;24(4):608-615.
- (4) Segerstolpe Å, Palasantza A, et al. Cell Metab. 2016 Oct 11; 24(4): 593-607.
- (5) Baron M, Veres A, et al. Cell Systems. 2016. Oct 26; 3(4): 346-360.
- (6) Lawlor N, George J, et al. Genome Res. 2017 Feb;27(2):208-222.
- (7) Enge M, Arda HE, et al. Cell. 2017 5; 171(2): 321-330.

2. Judging from the 10x web summary files, there exists considerable sample to sample variations. The authors claimed and showed that the sequencing data looked more uniform after filtering. Exactly how many cells were filtered out? What was the viability of input cells in each donor? Is viability correlated with the resulting single-cell sequencing profiles (for example, % of mitochondria reads?) To facilitate further evaluation of library quality and potential source of variation, in the supplementary figure 2, the author should consider include number of cells collected from each donor and number of cells passed QC. These parameters will help the science community to establish the standard of hepatocytes single-cell RNA-seq.

3. Page 8, line 172, some grammar error in the sentence "plasmablasts which traffic to the tissue..."

4. Page 10, line 209, "...respect to the zoned gene expression patterns previously shown in mouse (Fig 4d)". Does the author mean Figure S8?

5. The cell cycle phase prediction in Figure 2e, Figure 4a, Figure 5a, were not described in the main text.

Reviewer #3 (Remarks to the Author):

The authors have significantly strengthened their manuscript. While there are clearly some issues that pertain to the impact of experimental conditions on the observed heterogeneity, especially in the hepatocyte cell populations, the authors are very careful in pointing these out. Notably, human liver is extremely challenging to handle and a perfect 'clean' dataset would be very hard to obtain. Overall I feel this work represents an important and valuable resource for exploring cell states in the human liver. I have two minor comments:

- 1) Row 628 - Bahar Halpern et al found that 50% of hepatocyte genes are zoned (rather than liver genes, this work did not address other cell types).
- 2) Row 778 - Please provide the cell cycle marker genes in a supplementary table

Point-By-Point Responses to Reviewer Comments:

Reviewer 1

Thank you for your thoughtful and comprehensive responses to my criticisms. I am happy that my concerns have wherever practical been addressed. This is an important and extremely useful paper.

We sincerely thank the reviewer again for their very helpful suggestions.

Reviewer 2

In the revised manuscript, the authors included more in-depth single-cell sequencing QC matrices, reformatted some of the figures, compared the human hepatocytes single-cell RNA-seq data from landmark genes from mouse, and clarified some of the data interpretations. Overall, this results in a much-improved version of the manuscript.

I have the following remaining questions/comments:

- 1. Page 3, there are multiple single-cell studies on the human pancreas (Please note that all the references, including the one already cited in the manuscript, used isolated human islets as starting material). Please add in the following references:
(1) Li J, Klughammer J, et al. EMBO Rep. 2016 Feb;17(2):178-87.
(2) Wang YJ, Shug J, et al. Diabetes 2016 Oct; 65(10): 3028-3038.
(3) Xin Y, Kim J, et al. Cell Metab. 2016 Oct 11;24(4):608-615.
(4) Segerstolpe Å, Palasantza A, et al. Cell Metab. 2016 Oct 11; 24(4): 593–607.
(5) Baron M, Veres A, et al. Cell Systems. 2016. Oct 26; 3(4): 346-360.
(6) Lawlor N, George J, et al. Genome Res. 2017 Feb;27(2):208-222.
(7) Enge M, Arda HE, et al. Cell. 2017 5; 171(2): 321-330.*

We thank the reviewer for this suggestion, and have referenced each of these important papers in the revised manuscript on page 3, line **61**. We have also mentioned that the human pancreas work to date has only studied isolated islets on line **61** of the revised manuscript.

2. Judging from the 10x web summary files, there exists considerable sample to sample variations. The authors claimed and showed that the sequencing data looked more uniform after filtering. Exactly how many cells were filtered out? What was the viability of input cells in each donor? Is viability correlated with the resulting single-cell sequencing profiles (for example, % of mitochondria reads?) To facilitate further evaluation of library quality and potential source of variation, in the supplementary figure 2, the author should consider include number of cells collected from each donor and number of cells passed QC. These parameters will help the science community to establish the standard of hepatocytes single-cell RNA-seq.

The reviewer highlights the need for reporting additional QC statistics, which we agree is very important in the rapidly evolving field of single-cell RNA sequencing studies. We have included the requested information in a new Supplementary Figure 2A. We have also highlighted this the numbers of input cells that passed quality control in the text on Page 7 lines **141-143**.

3. Page 8, line 172, some grammar error in the sentence “plasmablasts which traffic to the tissue”

The grammar has been corrected in lines **174-176**.

4. Page 10, line 209, “respect to the zonated gene expression patterns previously shown in mouse (Fig 4d)”;. Does the author mean Figure S8?

We thank the reviewer for pointing this out. The reference to Supplementary Figure 8 was added on line **212**.

5. The cell cycle phase prediction in Figure 2e, Figure 4a, Figure 5a, were not described in the main text.

We apologize for this oversight. Figure 2e, Figure 4a & Figure 5a have been described in the legend of Figure 2 on lines **530 & 783** (2e), line **194** (4a) and line **324** (5a).

Reviewer 3

The authors have significantly strengthened their manuscript. While there are clearly some issues that pertain to the impact of experimental conditions on the observed heterogeneity, especially in the hepatocyte cell populations, the authors are very careful in pointing these out. Notably, human liver is extremely challenging to handle and a perfect 'clean' dataset would be very hard to obtain. Overall I feel this work represents an important and valuable resource for exploring cell states in the human liver. I have two minor comments:

We are pleased that the reviewer was content with our changes. We completely agree that the challenges of working with liver are not trivial and we are glad that this was clear from the manuscript. We hope that this data will help to contribute the overall understanding liver cellular biology.

1) Row 628 - Bahar Halpern et al found that 50% of hepatocyte genes are zonated (rather than liver genes, this work did not address other cell types).

Thank you for raising this important point- we have made the correction on Row **631**.

2) Row 778 - Please provide the cell cycle marker genes in a supplementary table

This information is now found in the new Supplementary Table 8, and referenced on line **783**.

REVIEWERS' COMMENTS:

Reviewer #2 (Remarks to the Author):

The authors have satisfactorily addressed all my questions.